# IFT cargo and motors associate sequentially with IFT trains to enter cilia of *C. elegans*

Aniruddha Mitra [1,2], Elizaveta Loseva [1] & Erwin J. G. Peterman [1] ✉

Intraflagellar transport (IFT) orchestrates entry of proteins into primary cilia. At the ciliary base, assembled IFT trains, driven by kinesin-2 motors, can transport cargo proteins into the cilium, across the crowded transition zone. How trains assemble at the base and how proteins associate with them is far from understood. Here, we use single-molecule imaging in the cilia of *C. elegans* chemosensory neurons to directly visualize the entry of kinesin-2 motors, kinesin-II and OSM-3, as well as anterograde cargo proteins, IFT dynein and tubulin. Single-particle tracking shows that IFT components associate with trains sequentially, both in time and space. Super-resolution maps of IFT components in wild-type and mutant worms reveal ciliary ultrastructure and show that kinesin-II is essential for axonemal organization. Finally, imaging cilia lacking kinesin-II and/or transition zone function uncovers the interplay of kinesin-II and OSM-3 in driving efficient transport of IFT trains across the transition zone.

Primary or sensory cilia protrude from most eukaryotic cells and are involved in detecting and relaying external cues to the cell body[1]. Ciliary function requires a tightly regulated pool of proteins in the cilioplasm and ciliary membrane, different in composition from the rest of the cell body. To maintain this heterogeneity, eukaryotic cells use a specialized intracellular transport system, intraflagellar transport (IFT; Fig. 1a)[2,3]. The core of a cilium is formed by an axoneme, nine doublet microtubules (MTs) in a cylindrical configuration, with the doublets emanating from the ciliary base. Located further along the cilium is a dense region called the transition zone (TZ, Fig. 1b) that acts as a diffusion barrier[4], with several ciliary proteins shown to hitch a ride along anterograde IFT trains to cross the TZ. IFT trains are large, ordered, polymeric structures with periodic repeats of IFT-B and IFT-A complexes[5–7]. Anterograde kinesin-2 motors associate with IFT-B proteins and drive the trains from the base, through the TZ, to the tip[3]. In *C. elegans*, anterograde IFT is driven by two kinesin-2 motors, heterotrimeric kinesin-II and homodimeric OSM-3. The slower kinesin-II navigates anterograde IFT trains across the TZ and is then gradually replaced by the faster OSM-3[8], which drives the trains to the ciliary tip[9,10].

While recent studies have revealed key aspects of the anterograde IFT mechanism, the overall picture of the dynamic processes occurring at the ciliary base is relatively incomplete. In fluorescence studies, we observed that pools of IFT-train, motor and cargo proteins are concentrated around the ciliary base, where anterograde IFT trains are assembled[8,11]. FRAP (fluorescence recovery after bleaching) experiments[12,13] and a recent structural study in *C. reinhardtii*[14] revealed that anterograde IFT trains are assembled sequentially at the ciliary base, with IFT-B complexes forming a template scaffold and IFT-A, IFT-dynein complexes and kinesin-II binding subsequently. Several ciliary membrane proteins have been shown to couple to anterograde IFT trains moving them across the TZ in a directed manner[15–17]. Cytosolic ciliary proteins, such as IFT dynein[18,19] and tubulin[20] also bind to anterograde IFT trains as cargo, though tubulin has also been shown to enter the cilium diffusively in *C. reinhardtii*[21,22]. While the structural features of IFT trains at the ciliary base have become clearer, the dynamics of how different ciliary proteins associate with IFT trains and navigate the TZ for ciliary entry have not yet been resolved.

In this study, we employ *C. elegans* chemosensory cilia as a model system to directly visualize how different proteins enter the cilium. Using small-window illumination microscopy (SWIM)[23], we image the entry of individual IFT motors (kinesin-II, OSM-3 and IFT-dynein) and tubulin into the cilia of the PHA/PHB neuron pair (Fig. 1a). Single-molecule analysis, along with numerical simulations, allows visualization of where and at what stage kinesin-2, IFT dynein and tubulin bind

[1]Department of Physics and Astronomy and LaserLaB, Vrije Universiteit Amsterdam, Amsterdam, The Netherlands. [2]Present address: Cell Biology, Neurobiology and Biophysics, Department of Biology, Faculty of Science, Utrecht University, Utrecht, The Netherlands. ✉e-mail: e.j.g.peterman@vu.nl

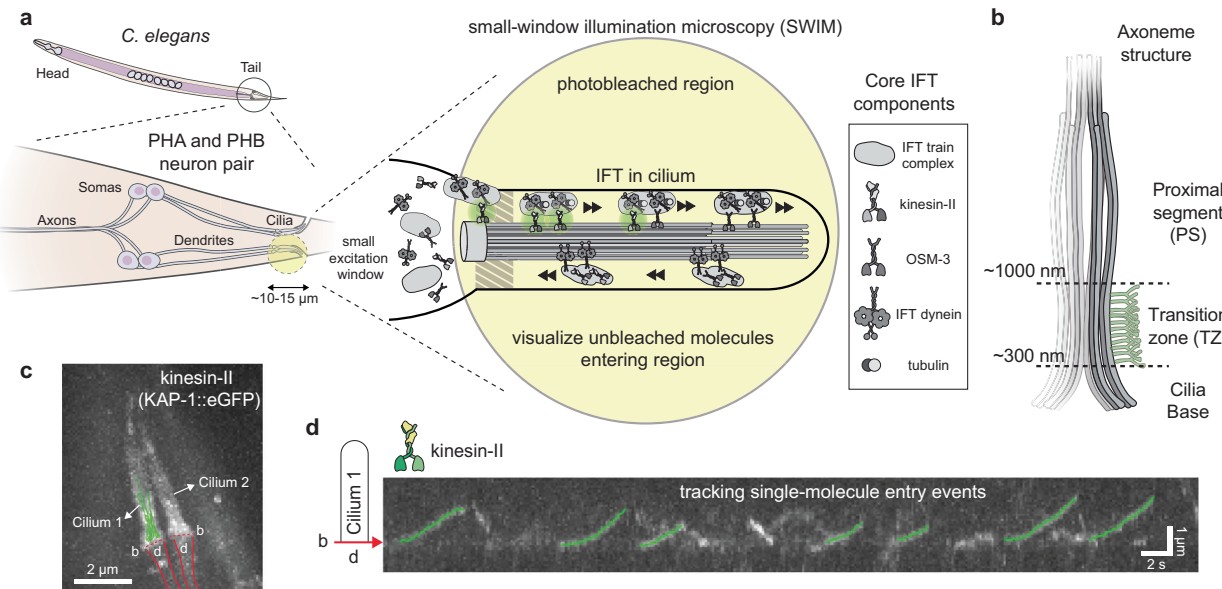

**Fig. 1 | Imaging approach to visualize individual kinesin-II (KAP-1::eGFP) molecules entering the cilia of PHA/PHB neurons. a** Illustration of single-molecule imaging of kinesin-II motors entering the sensory cilia of PHA/PHB neurons located in the tail of *C. elegans*, using small-window illumination microscopy (SWIM). Using SWIM, we can visualize not-yet-photobleached kinesin-II molecules in a small excitation window (~10–15 μm) for a long duration of time. **b** Illustration of the axonemal structure at initial part of the cilia. The axoneme starts at ciliary base, passes through the TZ between ~300–1000 nm, moving into the proximal segment. **c** Maximum projection of eGFP-labeled kinesin-II (KAP-1::eGFP) imaged in PHA/PHB cilia of a worm (see Supplementary Movie 2a). Tracks of single-molecule events entering cilium 1 have been indicated (green lines), b indicates the base of the two cilia (dotted red lines) and d indicates the region where the corresponding dendrites are located (between the solid red lines). **d** Representative kymograph (space-time intensity plot) of kinesin-II along cilium 1, in (**c**), displays that single kinesin-II molecules, diffusing in the dendrite, dock at the ciliary base and enter the cilium. Single-molecule entry events are tracked (tracks indicated in green).

to assembling and moving anterograde IFT trains. Furthermore, single-molecule localizations provide insight into ultrastructure and organization at the proximal part of the cilium, highlighting the role played by kinesin-II in maintaining ciliary structure. Comparison of wild-type worms and mutants lacking kinesin-2 motor or TZ function provides a detailed picture of the nature of the TZ and the roles played by the two kinesin-2 motors in navigating anterograde IFT trains across the TZ.

## Results

### Single-molecule imaging reveals that kinesin-II pauses briefly before entering the cilium

To visualize the dynamics of individual proteins entering cilia, we performed SWIM on fluorescently labeled proteins in the chemosensory neurons in *C. elegans*[23]. The idea of SWIM is to excite and photobleach fluorophores only within a small region of the sample (diameter 10–15 μm), using a high laser intensity, allowing continuous entry of "fresh" proteins into the excited region. Here, we only illuminated the cilia (~8 μm long), the periciliary membrane compartment (PCMC; at the transition between dendrite and cilia; ~1 μm) and small sections of the dendrites (~1–2 μm) of a PHA/PHB neuron pair located near the tail of *C. elegans* (Fig. 1a). Compared to standard wide-field illumination, SWIM allows imaging of single-molecule events for a much longer duration (>20 mins; Supplementary Movie 1 and Supplementary Fig. 1), with a substantially higher signal-to-background ratio, because of the reduced out-of-focus autofluorescence background[23].

To study ciliary entry dynamics, we first visualized single kinesin-II motors (KAP-1::eGFP, non-motor subunit of kinesin-II; Supplementary Movie 2a and Fig. 1c). We observed that kinesin-II motors show diffusive behavior in the dendrite and the PCMC, likely in an autoinhibited state[24] not interacting with dendritic microtubules. At random times, individual motors become immobile, close to the ciliary base (Fig. 1d), indicated by the stochastic appearance of a fluorescent punctum at the ciliary base (Supplementary Movie 2a). Typically, motors remain immobile for a while, before starting to move along the axoneme, speeding up after

crossing the TZ. We hypothesized that kinesin-II associates with or "docks" to immobile, assembling IFT trains[14], which releases the motor autoinhibition, allowing them to drive the trains into the cilium. To obtain a quantitative picture of the dynamics we extracted single-molecule tracks from the image sequences. Molecules were selected for tracking only when a single molecule docked at the ciliary base, with no other molecules docking in the vicinity. In order to pool information from multiple cilia in different worms, we transformed the track coordinates $(x_i, y_i)$ to ciliary coordinates $(c_{\parallel\_i}, c_{\perp\_i}$, Fig. 2a and Supplementary Fig. 2). The 311 tracks show that kinesin-II primarily docks to IFT trains between 0–400 nm from the ciliary base (Fig. 2b, c), on average at $188 \pm 20$ nm. Most of the motors walk less than 2000 nm into the cilia, before photobleaching or detaching from the IFT train and axoneme[8,25]. From the single-molecule tracks, we calculated the average point-to-point velocity along the ciliary length and observed that it increases gradually from ~100 nm/s to 350 nm/s across the TZ, before increasing substantially after ~1 μm, where the TZ ends (Fig. 2d). For each track we also determined the (measured) pause time, $t_{p\_m}$, which we defined as the time it takes for a kinesin-II to move 100 nm along the cilium (Fig. 2e). The average pause time is $0.9 \pm 0.1$ s, with larger pause times (>1.5 s) mostly observed for events docking before 400 nm from the start of the cilium (Fig. 2f). In our experiments eGFP-labeled kinesin-II photobleaches readily and we only observe motors that have not photobleached. To take into account the effect of photobleaching we simulated entry events (see Supplementary Fig. 3a and Methods). For each simulation condition, we provided a characteristic bleach time, $t_{bleach}$ (obtained experimentally; 1.6 s for kinesin-II; Supplementary Fig. 3b), and an "actual" pause time, $t_{p\_actual}$, and determined the measured pause time, $t_{p\_m}$. Parameters could be found ($t_{p\_actual} = 0.7$ s, $N = 1500$, $N_{unbleached} = 320$, $t_{p\_m} = 0.9 \pm 0.1$ s), yielding simulated distributions similar to the experimental ones (Fig. 2g). The simulations reveal that $t_{p\_m}$ increases linearly with $t_{p\_actual}$ before saturating at $\sim t_{bleach}$ (Fig. 2h). Thus, our determination of pause times is affected by photobleaching when $t_{p\_actual}$ is similar to (or longer than) $t_{bleach}$ of the

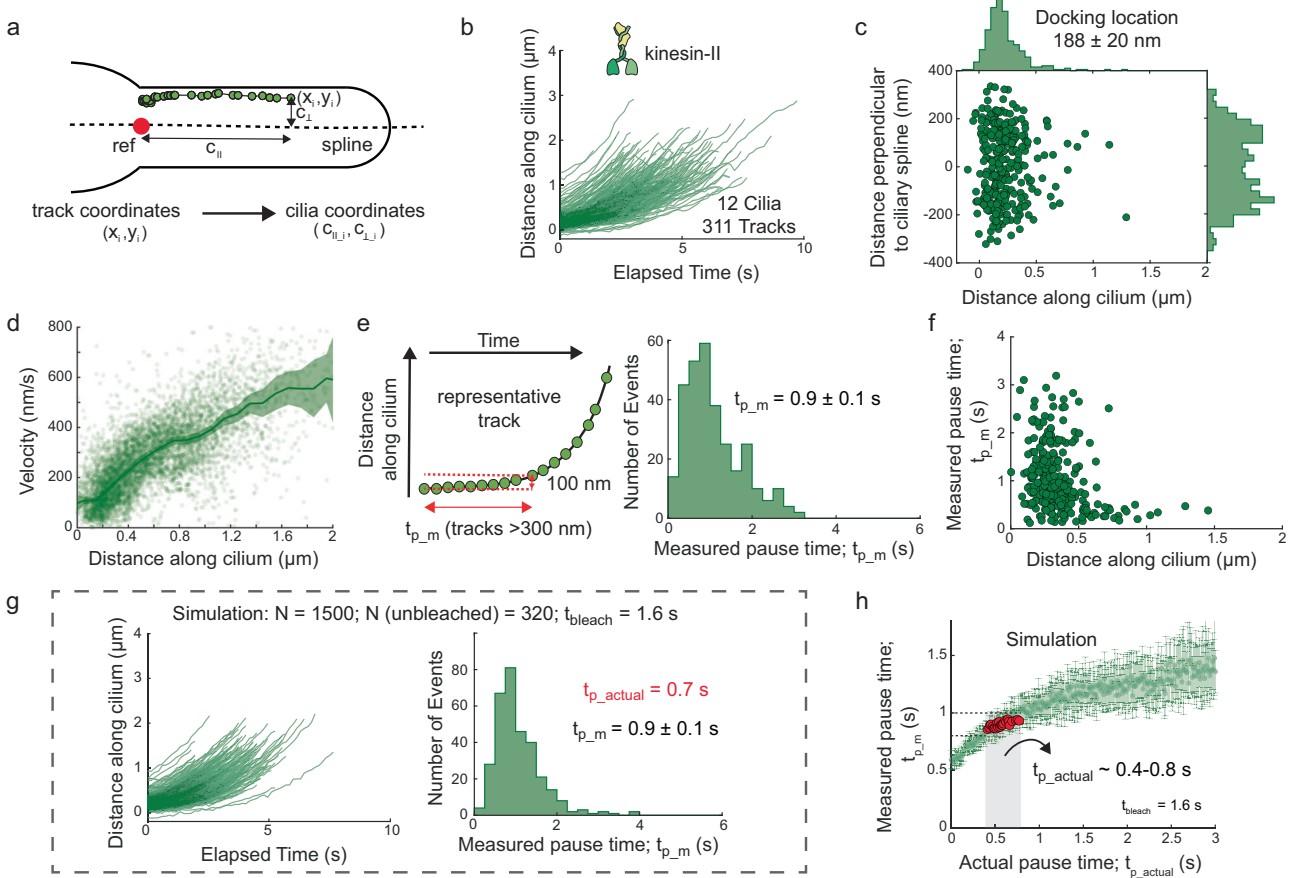

**Fig. 2 | Analysis of the single-molecule tracks of kinesin-II (KAP-1::eGFP) molecules entering cilia. a** By drawing a central spline along the cilium and defining a reference point (at ciliary base), x-y coordinates of tracks are transformed to ciliary coordinates, as shown in the scheme (also see Supplementary Fig. 2). **b** Distance-time plots of all kinesin-2 tracks pooled together (311 tracks from 12 cilia). **c** Docking locations of all kinesin-2 tracks at ciliary base. Distributions of docking locations along ciliary length (upper histogram) and perpendicular to ciliary spline (right histogram). Kinesin-II molecules mostly dock between 0–0.5 μm from ciliary base with average docking location at 188 ± 20 nm. **d** Distribution of point-to-point velocity along cilium length (6693 datapoints). Solid line represents binned average velocity and shaded area indicates error. **e** Left: Measured pause time, $t_{p\_m}$ is defined as amount of time it takes for a track to move 100 nm, with only tracks longer than 300 nm included in the analysis. Right: Histogram of measured pause time (average pause time $t_{p\_m}$ 0.9 ± 0.1 s). **f** Correlation of $t_{p\_m}$ with docking

location of kinesin-II events. **g** Distance-time plots of simulated kinesin-II entry events (left panel) and histogram of measured pause time (average $t_{p\_m}$ 0.9 ± 0.1 s; right panel), with actual pause time, $t_{p\_actual}$, 0.7 s and characteristic bleach time, $t_{bleach}$, 1.6 s (as calculated in Supplementary Fig. 3b). Out of the 1500 simulated events ($N$) only 320 do not bleach before moving 300 nm from docked location. Simulation details are provided in Supplementary Fig. 3. **h** Distribution of $t_{p\_m}$ with respect to $t_{p\_actual}$, obtained from numerical simulations of kinesin-II molecules entering cilia, assuming $t_{bleach}$ of 1.6 s (from experiments; Supplementary Fig. 3b). Each point represents the average $t_{p\_m}$ for a single simulated experiment of 1500 events (example simulation in 2 g), varying the parameter $t_{p\_actual}$ by steps of 0.01 s. Simulated experiments yielding $t_{p\_m}$ 0.9 ± 0.1 s (similar to experimental value; **e**) are highlighted in red, indicating that $t_{p\_actual}$ is in the range 0.4–0.8 s. Average value and error are estimated using bootstrapping (see "Methods").

fluorescent label. For kinesin-II, we estimate $t_{p\_actual}$ to be 0.4–0.8 s. In summary, single-molecule analysis, complemented by simulations, allows a quantitative description of how single kinesin-II motors associate with anterograde IFT trains, pause briefly, and enter the cilium.

## IFT dynein pauses at the ciliary base while OSM-3 and tubulin enter the cilium without a clear pause

Next, we used this single-molecule imaging and analysis approach to uncover how OSM-3 (OSM-3::mCherry), IFT dynein (XBX-1::eGFP) and tubulin dimers (TBB-4::eGFP; β-tubulin isoform[26]) enter cilia (Supplementary Movie 2). Like kinesin-II, for each of these proteins we observed a freely diffusing pool in the PCMC, with individual proteins stochastically docking to trains at the ciliary base or deeper inside the cilium before moving into the cilium. To our surprise, we found that the distributions of docking locations and pause times are remarkably different for the different proteins. In comparison to kinesin-II, OSM-3 attaches to trains slightly deeper into the TZ (Fig. 3a, b), pausing much shorter than kinesin-II or not at all (Supplementary Fig. 4a and

Supplementary Movie 2b). Numerical simulations indicate that the $t_{p\_actual}$ for OSM-3 ranges from 0 to 0.1 s (Fig. 3c and Supplementary Fig. 3c). This indicates that, unlike kinesin-II, OSM-3 primarily attaches to already moving trains, without pausing. For IFT dynein we observed that it attaches within a very narrow region at the ciliary base (Fig. 3d, e) but pauses much longer than kinesin-2, with an average $t_{p\_m}$ of 1.8 ± 0.2 s, with some motors pausing >5 s (Supplementary Fig. 4b and Supplementary Movie 2c). Simulations show that $t_{p\_actual}$ of IFT-dynein mostly ranges between 2.7 to 8 s (Fig. 3f and Supplementary Fig. 3d), which is substantially longer than kinesin-II. For single-molecule imaging of tubulin, prolonged exposure to high intensity 491 nm laser was necessary to photobleach axoneme-lattice-incorporated tubulin. Only then "new" tubulin molecules could be observed to diffuse, to move in a directed manner and/or to be incorporated into the axoneme lattice (Supplementary Movie 2d). We observed that single tubulin molecules dock onto moving anterograde trains at the ciliary base, throughout the TZ as well as the proximal segment (PS; Fig. 3g, h). Tubulin appears to dock only to moving trains,

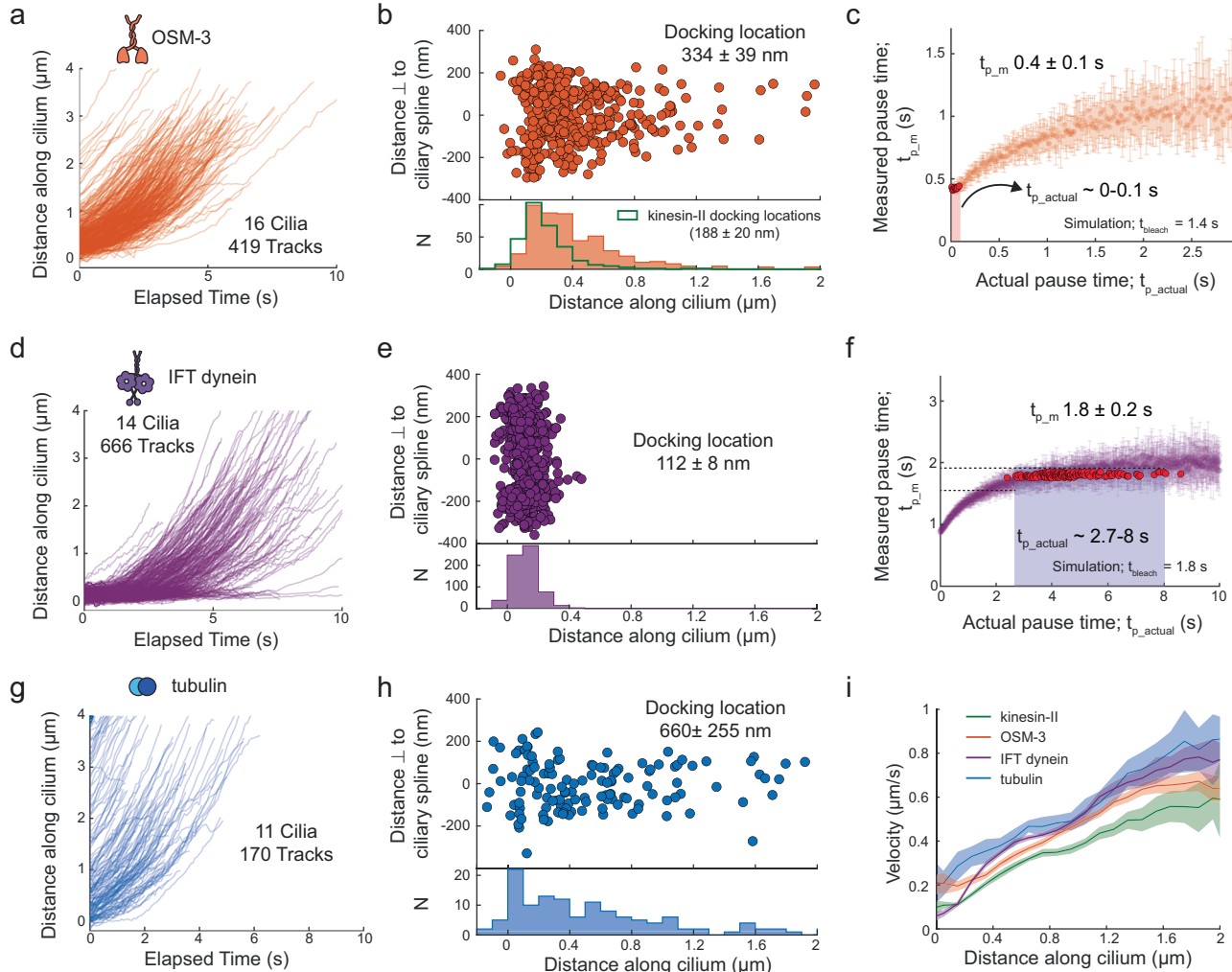

**Fig. 3 | Dynamics of single OSM-3 (OSM-3::mCherry), IFT dynein (XBX-1:eGFP) and tubulin (TBB-4::eGFP) molecules entering cilia. a** Distance-time plots of OSM-3 (419 tracks from 16 cilia). **b** Distribution (upper panel) and histogram (lower panel) of docking locations of OSM-3 (average docking location is 334 ± 39 nm). The distribution of docking locations of kinesin-II is overlayed (green line). **c** Distribution of the measured pause time, $t_{p\_m}$, with respect to actual pause time, $t_{p\_actual}$, for OSM-3, obtained from numerical simulations (using $t_{bleach}$ 1.4 s). Each point represents the average $t_{p\_m}$ for a single simulated experiment of 1500 events (example simulation in Supplementary Fig. 3c), varying $t_{p\_actual}$ by steps of 0.01 s. Simulated experiments yielding $t_{p\_m}$ 0.4 ± 0.1 s (similar to experimental value; Supplementary Fig. 4a) are highlighted in red, indicating that $t_{p\_actual}$ is in the range 0–0.1 s. **d** Distance-time plots of IFT dynein (666 tracks from 14 cilia). **e** Distribution (upper panel) and histogram (lower panel) of docking locations of IFT dynein

(average docking location is 112 ± 8 nm). **f** Distribution of the measured pause times, $t_{p\_m}$, with respect to actual pause times, $t_{p\_actual}$, for IFT dynein, obtained from numerical simulations (using $t_{bleach}$ 1.8 s). Each point represents the average $t_{p\_m}$ for a single simulated experiment of 10,000 events (example simulation in Supplementary Fig. 3d), varying $t_{p\_actual}$ by steps of 0.01 s. Simulated experiments yielding $t_{p\_m}$ 1.8 ± 0.2 s (similar to experimental value; Supplementary Fig. 4b) are highlighted in red, indicating that $t_{p\_actual}$ is in the range 2.7–8 s. **g** Distance-time plots of tubulin (170 tracks from 11 cilia). **h** Distribution (upper panel) and histogram (lower panel) of docking locations of tubulin (average docking location is 660 ± 255 nm). **i** Distribution of binned average velocities (solid line) along the length of the cilium for kinesin-II (green), OSM-3 (orange), IFT dynein (purple), tubulin (blue), with the shaded areas indicating the errors. Average values and errors are estimated using bootstrapping (see "Methods").

deeper inside the cilium, causing the $t_{p\_m}$ (Supplementary Fig. 4c) and $t_{p\_actual}$ (Supplementary Figs. 3e and 4d) to be negligible. Overall, we find that tubulin, and to a lesser extent, OSM-3, docks onto already moving anterograde trains throughout the proximal part of the cilium, indicating that both are also capable of diffusing through the TZ. In contrast, IFT dynein associates with anterograde IFT trains assembling at the ciliary base, pausing for several seconds, before being carried across the TZ as passive cargo.

## Kinesin-II switches on and off IFT trains resulting in a lower apparent velocity

Furthermore, we determined the velocities of the different IFT components along the ciliary proximal part and observed striking differences (Fig. 3i). Kinesin-II velocity is lower than OSM-3, throughout TZ and PS. IFT-dynein velocity is low near the base, but increases in the TZ,

becoming higher than both kinesin-II and OSM-3. Furthermore, tubulin moves significantly faster than both anterograde kinesin-2 motors all along the cilium. Thus, we observed that the velocity readout is different for different IFT components even though they likely associate with a similar pool of anterograde IFT trains. To obtain a higher time-resolution picture of IFT motors moving across the TZ, we imaged them at a ~5-10x faster frame rate (Supplementary Movie 3). Using this faster imaging we could clearly resolve single diffusive IFT motors docking at the ciliary base. Furthermore, for kinesin-II, we observed that while some molecules docked and moved unidirectionally into the cilium (molecule 1 in Supplementary Fig. 4e and Supplementary Movie 3a), others showed short diffusive bursts in between bouts of directional motion into the cilium (molecule 2 in Supplementary Fig. 4e and Supplementary Movie 3a). On rare occasions, we also observed kinesin-II motors docking and undocking from stationary trains, with diffusive bursts in

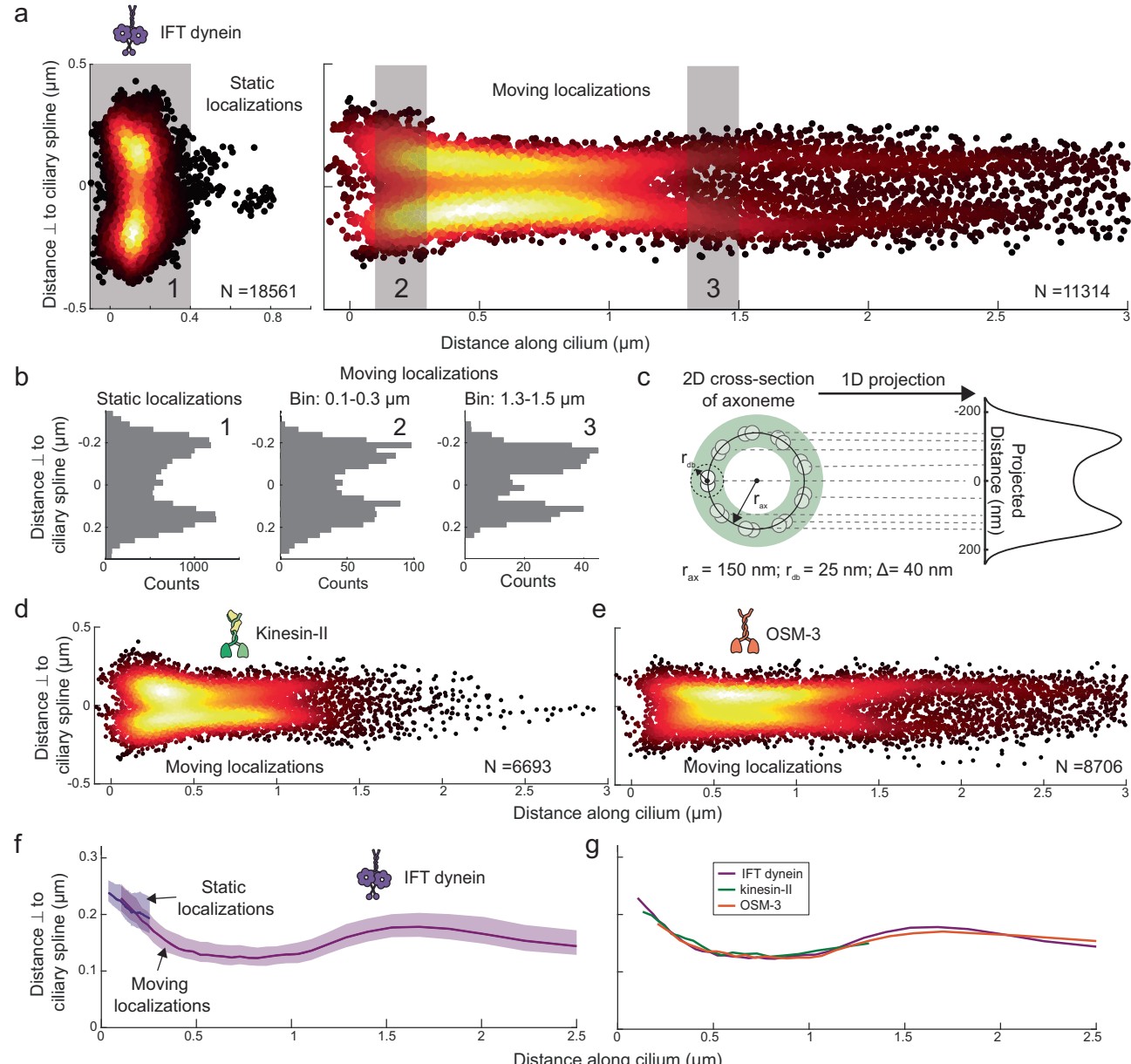

**Fig. 4 | Single-molecule localizations of IFT dynein, kinesin-II and OSM-3 reveal the structure at the proximal part of cilia. a** Super-resolution map of single-molecule localizations obtained from 666 IFT-dynein tracks. Left panel: $N = 18{,}561$ static localizations. Right panel: $N = 11{,}314$ moving localizations. **b** The distribution of distances perpendicular to the spline for static localizations (1), and of moving localizations between 0.1–0.3 μm (2) and between 1.3–1.5 μm (3); indicated with light gray rectangles in (**a**). **c** Illustration of the transverse section of the hollow axoneme (radius $r_{ax}$) comprising of 9 microtubule doublets, with IFT components moving along individual doublets within the radius $r_{db}$. Right panel: 2D projection of this 3D geometry ($N = 10{,}000$; $r_{ax} = 150$ nm, $r_{db} = 25$ nm and localization error $\Delta = 40$ nm). Super-resolution map of single-molecule localizations of 311 kinesin-II

tracks (**d**; $N = 6693$ moving localizations) and 419 OSM-3 tracks (**e**; $N = 8706$ moving localizations). **f** Width of the distribution of single-molecule localizations with respect to the ciliary spline, along the ciliary length for IFT dynein (static localizations shown in purple and moving localizations in magenta; also indicated by arrows). To estimate width of distribution, the 80th percentile value of the cumulative distribution of perpendicular distance, for every sampled bin of 1000 localizations, is plotted. The 70–90th percentile range indicated by the shaded area provides a measure of the error. **g** Width of the distribution of single-molecule localizations with respect to the ciliary spline, along the ciliary length for IFT dynein (magenta), kinesin-II (green), OSM-3 (orange).

between (Supplementary Fig. 4f and Supplementary Movie 3b). In contrast, IFT-dynein and OSM-3 molecules primarily docked and crossed the TZ while moving unidirectionally (Supplementary Fig. 4g, h and Supplementary Movie 3c, d). This suggests that, as reported previously[25], kinesin-II motors can stochastically detach from IFT trains, diffuse and bind again to the same or a different train within a short period of time. These short diffusive bursts are likely missed when imaging at a lower frame rate and result in the lower apparent velocity observed for kinesin-II, and to a lesser extent OSM-3. Our observations

suggest that kinesin-2 has a lower average velocity, because it stochastically detaches and attaches to IFT trains. In contrast, IFT-dynein and tubulin remain more tightly associated with IFT trains and consequently move with an average velocity close to that of the trains.

## Single-molecule localizations allow mapping of ciliary ultrastructure
To visualize the architecture of the ciliary base, we created super-resolution maps of the ciliary components from the tracked single-

**Table 1 | Data obtained from tracking single molecule events of different IFT components, entering the phasmid cilia**

| Worm strain | No. of cilia (No. of worms) | No. of tracks | Moving localizations (Paused localizations) | Docking location (nm) | Measured pause time; $t_{p\_m}$(s) (No. of tracks) | Estimated real pause times; $t_{p\_actual}$(s) |
|---|---|---|---|---|---|---|
| kinesin-II | 12 (7) | 311 | 6693 (225) | 188 ± 20 | 0.9 ± 0.1 (291) | 0.4–0.8 |
| OSM-3 | 16 (9) | 419 | 8706 (88) | 334 ± 39 | 0.4 ± 0.1 (415) | 0–0.1 |
| IFT-dynein | 14 (7) | 666 | 11,314 (18,561) | 112 ± 8 | 1.8 ± 0.2 (263) | 2.7–8 |
| Tubulin | 11 (7) | 170 | 4085 (36) | 660 ± 255 | 0.2 ± 0 (169) | 0 |
| OSM-3 *kap-1* | 14 (8) | 304 | 8085 (1471) | 315 ± 72 | 0.7 ± 0.1 (286) | 0.4–0.8 |
| IFT-dynein *kap-1* | 28 (17) | 420 | 4974 (6859) | 189 ± 25 | 1.4 ± 0.3 (261) | 1.5–3.5[a] |
| kinesin-II *osm-3* | 20 (10) | 299 | 9490 (603) | 229 ± 30 | 0.7 ± 0.1 (274) | 0.4–0.8 |
| OSM-3 *mksr-1* | 14 (9) | 429 | 11,672 (116) | 428 ± 56 | 0.3 ± 0 (428) | 0[a] |
| kinesin-II *mksr-1* | 12 (9) | 112 | 1741 (148) | 186 ± 29 | 0.7 ± 0.2 (102) | 0.3–0.5[a] |
| OSM-3 *kap-1; mksr-1* | 10 (7) | 191 | 3923 (1137) | 230 ± 56 | 0.9 ± 0.2 (173) | 1.1–1.9 |

For measured pause time information only tracks moving >300 nm were considered (this number is indicated).
[a]indicates that in these cases none of the simulations yield the experimentally obtained $t_{p\_m}$, hence the closest $t_{p\_m}$ is used to provide an estimate.

molecule localizations. We obtained 30,977 IFT-dynein localizations, which we classified into "static" (18,561) and "moving" localizations (11,314) using a windowed mean-squared-displacement (MSD) approach[17,25]. The static IFT-dynein localizations observed between 0 and 400 nm along the cilium, with a peak at ~100 nm (Fig. 4a) and a width of ~500 nm (Fig. 4b), highlight the region of the ciliary base where IFT dynein docks onto assembling anterograde IFT trains. The moving IFT-dynein localizations reveal the shape of the initial part of the cilium (Fig. 4a). The distribution of localizations perpendicular to the ciliary spline is relatively broad at the ciliary base (~500 nm wide at 0.2 μm), tapers at the TZ (~250 nm wide between 0.4–1.2 μm) and "bulges" again at the ciliary proximal segment (~400 nm wide >1.5 μm). The distributions of both static and moving localizations appear hollow, with substantially more localizations along the periphery (Fig. 4b). Since the axoneme is a hollow cylinder composed of 9 microtubule doublets, we propose that the observed distribution arises from the 2D projection (by our imaging approach) of a 3D hollow cylindrical distribution. To confirm this, we simulated localizations along the 2D cross-section of the axoneme, with input parameters, radius of the axoneme ($r_{ax}$), radius of localizations around microtubule doublets ($r_{db}$), and localization error (Δ; Fig. 4c). The projection of these localizations is bimodal, centered around zero (Fig. 4c), similar to the experiments. We tried to estimate the underlying shape and width of the rim of the cylinder ($r_{db}$) along which IFT components are distributed by mapping the localization information to our simulations. We found that while we can precisely determine the cylinder diameter (Fig. 4f), we cannot accurately estimate $r_{db}$, due to experimental limitations, such as localization error and limited depth of field (Supplementary Fig. 5 and Supplementary text). Next, we classified the single-molecule localizations of kinesin-II and OSM-3 and obtained a negligible number of static localizations (Table 1). Super-resolution maps of moving kinesin-II (Fig. 4d) and OSM-3 localizations (Fig. 4e) are similar to IFT dynein (Fig. 4a), suggesting that the underlying 3D distribution is the same (Fig. 4g). In summary, super-resolution maps provided by single-molecule localizations allow accurate determination of the cylindrical shape of the ciliary proximal part, governed by the axonemal MT doublets.

### Entry dynamics of IFT components and cilia structure is altered in kinesin-II loss-of-function mutants

To obtain insight into the roles of kinesin-II and OSM-3 at the ciliary base, we imaged entry of IFT motors in mutant worms lacking OSM-3 (*osm-3*) or kinesin-II (*kap-1*) function (Supplementary Movie 4). In the considerably shorter[10] cilia of *osm-3* mutant worms, we observed that the pausing behavior of kinesin-II motors entering the cilium is similar

to wild type (Supplementary Fig. 6a), although the average docking location is marginally deeper into the cilium (Supplementary Fig. 6b). The velocity is also slightly lower in the TZ (Supplementary Fig. 6c), due to absence of faster OSM-3 motors. In *kap-1* mutant worms, with ciliary length similar to wild type, however, the velocity of IFT components as well as ciliary shape are very different[8,27]. We observed that OSM-3 in *kap-1* worms pauses significantly longer at the ciliary base (Fig. 5a and Supplementary Fig. 6d), with $t_{p\_actual}$ estimated to be in the range 0.4–0.8 s (Fig. 5c). In comparison to OSM-3 in wild-type worms, velocity of OSM-3 *kap-1* is much lower in the initial part of the TZ but increases faster, to a substantially higher velocity when the motors enter the PS (Fig. 5d). The distribution of docking locations is similar to wild type, although the number of dockings is higher at the ciliary base (in the bin between 0–100 nm; Fig. 5b), where OSM-3 takes over the role of absent kinesin-II. Remarkably, we observed that IFT-dynein motors in *kap-1* worms dock to IFT trains at the ciliary base as well as deeper inside the TZ (Fig. 5e, f), with a velocity distribution similar to OSM-3 *kap-1* (Supplementary Fig. 6e). Furthermore, the distribution of $t_{p\_m}$ and the estimated $t_{p\_actual}$ is lower than for IFT dynein in wild-type worms, with a large fraction of IFT dynein *kap-1* pausing only briefly (Supplementary Fig. 6f, g). This indicates that in *kap-1* worms, either IFT-dynein motors can dock to already assembled IFT trains that are slowly moving in the TZ and/or IFT-train assembly takes less time and assembly also takes place in the initial part of the TZ.

Super-resolution maps generated from single-molecule localizations of OSM-3 *kap-1* and IFT-dynein *kap-1* provide important insights in the ciliary structure of *kap-1* worms. First, the fraction of static localizations is significantly higher for OSM-3 in *kap-1* worms (Table 1), localizing in the initial part of the TZ, as well as at the ciliary base (Fig. 5g upper left). Second, the distribution of static localizations perpendicular to the ciliary spline is narrower for IFT-dynein *kap-1* in comparison to wild type (Fig. 5h), suggesting that the organization at the base is altered. Third, the super-resolution maps of moving localizations of OSM-3 *kap-1* and IFT-dynein *kap-1* show that, in contrast to wild-type cilia, the ciliary bulge distal from the TZ is absent or far less pronounced in *kap-1* mutant worms (Fig. 5i). Furthermore, the localization distributions at the proximal segment (>1.5 μm) do not show two clear maxima equidistant from the ciliary spline (Fig. 5g), as in wild type (Fig. 4a, e). This indicates that, in *kap-1* mutants, the axonemal structure might be compromised at the proximal segment, with randomly distributed doublets providing a less well-defined hollow space in the center. In summary, we find that in kinesin-II loss-of-function mutants, OSM-3 can partially take over the role of kinesin-II at the ciliary base, but IFT-train assembly is affected: assembly appears to be less localized and take less time. Furthermore, the axoneme

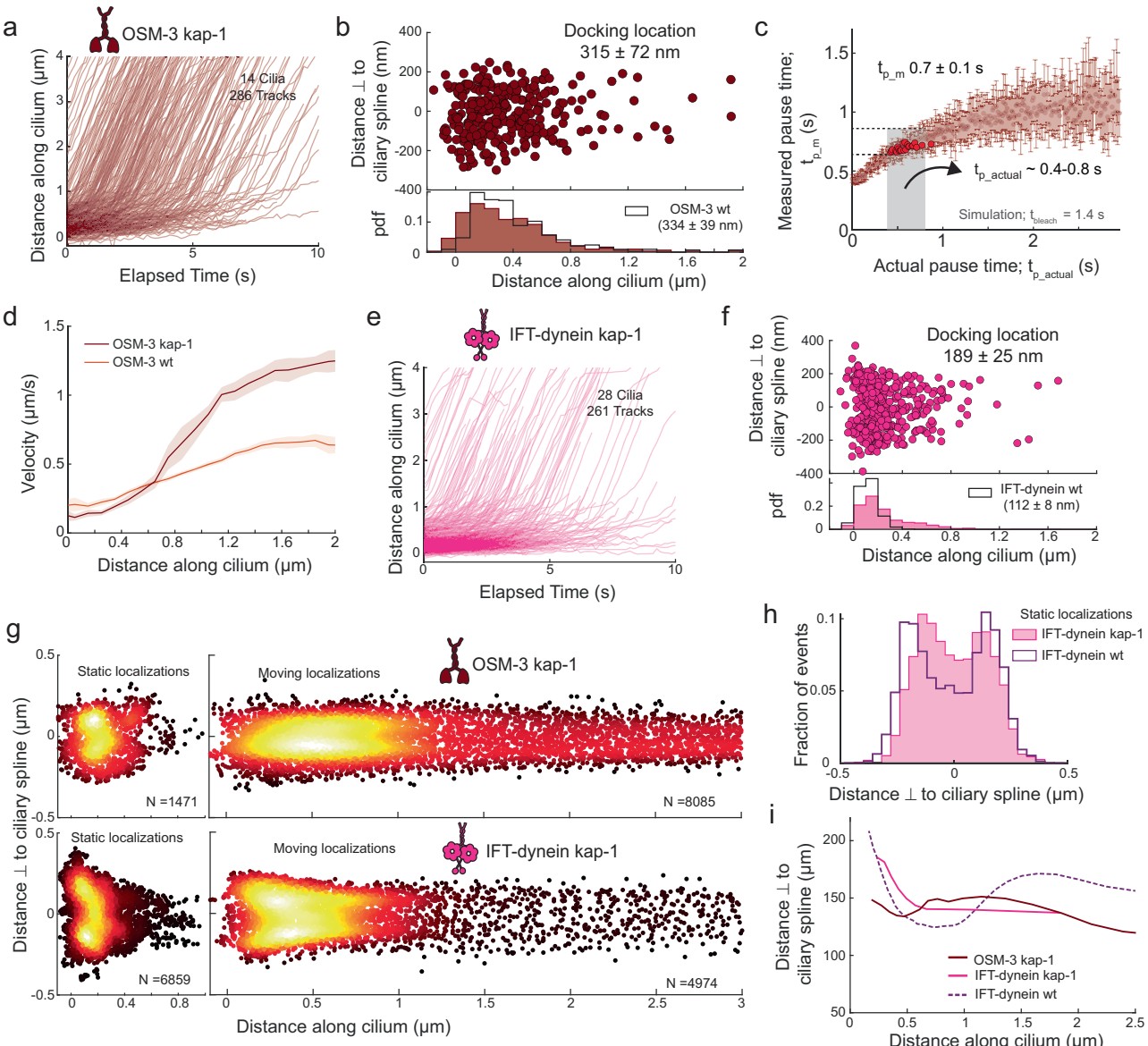

**Fig. 5 | Entry dynamics of OSM-3 and IFT-dynein molecules in cilia of kinesin-II loss-of-function mutants (*kap-1*). a** Distance-time plots of OSM-3 *kap-1* (286 tracks from 14 cilia). **b** Distribution (upper panel) and histogram (lower panel) of docking locations of OSM-3 *kap-1* (brown; average docking location is 315 ± 72 nm). Distribution of docking locations of OSM-3 in wild type is overlaid (black line). **c** Distribution of measured pause times, $t_{p\_m}$, with respect to actual pause times, $t_{p\_actual}$, for OSM-3 *kap-1* obtained from numerical simulations (using $t_{bleach}$ = 1.4 s). Each point represents the average $t_{p\_m}$ for a single simulated experiment of 1500 events, varying $t_{p\_actual}$ by steps of 0.01 s. Simulated experiments yielding $t_{p\_m}$ 0.7 ± 0.1 s (similar to experimental value; Supplementary Fig. 6d) are highlighted in red, indicating that $t_{p\_actual}$ is in the range 0.4–0.8 s. **d** Velocity distribution of OSM-3 along the ciliary length in wild-type (orange) and *kap-1* mutant (brown) worms. Solid line is binned average velocity and shaded area indicates the error. **e** Distance-time plots of IFT-dynein *kap-1* (261 tracks from 28 cilia). **f** Distribution (upper panel) and histogram (lower panel) of docking locations of IFT-dynein *kap-1* (pink; average

docking location is 189 ± 25 nm). Distribution of docking location of IFT-dynein in wild type is overlaid (black line). **g** Super-resolution map of single-molecule localizations corresponding to OSM-3 *kap-1* (upper panels) and IFT-dynein *kap-1* tracks (lower panels). OSM-3 *kap-1* has 1471 static localizations (upper left panel) and 8085 moving localizations (upper right panel). IFT-dynein *kap-1* has 6859 static localizations (lower left panel) and 4974 moving localizations (lower right panel). **h** Distribution of distances perpendicular to ciliary spline for static localizations of IFT-dynein *kap-1* (pink filled) and IFT-dynein in wild type (purple outline). **i** Width of distribution of single-molecule localizations with respect to the ciliary spline, along the ciliary length, for IFT-dynein in wild type (dotted purple), IFT-dynein *kap-1* (pink) and OSM-3 *kap-1* (brown). To estimate width of distribution, 80th percentile value of the cumulative distribution of perpendicular distance, for every sampled bin of 1000 localizations, is plotted. Average values and errors are estimated using bootstrapping.

structure is altered, indicating that kinesin-II plays a crucial role in building the ciliary architecture.

## IFT motors move more readily into the cilium when TZ is disrupted

Finally, we explored the entry dynamics of kinesin-2 in MKS-1-related protein 1 mutant worms (*mksr-1*), in which the organization and

integrity of the TZ is disrupted[8,28] (Supplementary Movie 5). The distribution of the docking locations of OSM-3 in *mksr-1* worms extends slightly deeper into the TZ and the PS (Fig. 6a, b), suggesting that OSM-3 may enter the cilium more readily when the TZ is compromised. OSM-3 (Fig. 6f) and kinesin-II velocities (Fig. 6c) are higher in *mksr-1* worms, throughout the proximal part of the cilium. In *kap-1; mksr-1* double-mutant worms, OSM-3 primarily docks at the ciliary base

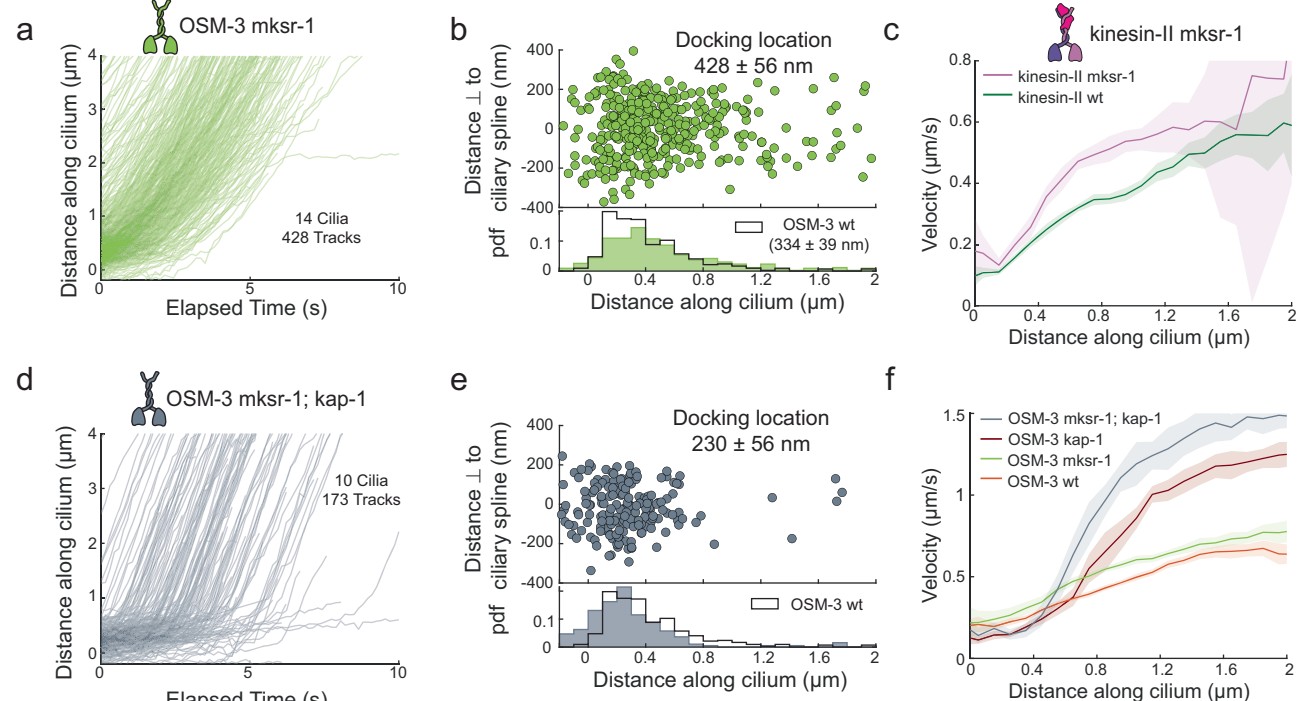

**Fig. 6 | Entry dynamics of OSM-3 and kinesin-II molecules in cilia of *mksr-1* mutants and *kap-1*; *mksr-1* double mutants. a** Distance-time plots of OSM-3 *mksr-1* (428 tracks from 14 cilia). **b** Distribution (upper panel) and histogram (lower panel) of docking locations of OSM-3 *mksr-1* (average docking location is 428 ± 56 nm). The distribution of docking locations of OSM-3 in wild-type worms is overlayed in black. **c** Velocity distribution of kinesin-II in wild type (green) and kinesin-II *mksr-1* (pink) along the ciliary length. Solid line is the binned average velocity and shaded area indicates the error. **d** Distance-time plots of OSM-3 *kap-1*; *mksr-1* (173 tracks from 10 cilia). **e** Distribution (upper panel) and histogram (lower panel) of docking locations of OSM-3 *kap-1*; *mksr-1* (gray; average docking location is 230 ± 56 nm). The distribution of docking locations of OSM-3 in wild-type worms is overlayed in black. **f** Velocity distribution of OSM-3 in wild type (orange), OSM-3 *kap-1* (brown), OSM-3 *mksr-1* (light green) and OSM-3 *kap-1*; *mksr-1* (gray) along the ciliary length. Solid line is the binned average velocity and shaded area indicates the error. Average value and error are estimated using bootstrapping (also in **c**).

(Fig. 6d, e), moving slowly through the initial part of the TZ before accelerating beyond -0.5 μm (Fig. 6f). In comparison to *kap-1*, the velocity increase occurs earlier in the cilium and the velocity is considerably higher, even in the initial part of the proximal segment. In both *mksr-1* and *kap-1*; *mksr-1* worms, the higher velocity indicates that the motion of the IFT trains is less hindered and/or more kinesin-2 motors are engaged with the trains in the absence of an intact TZ, resulting in faster trains. Furthermore, in FRAP (fluorescence recovery after photobleaching) experiments of OSM-3 in wild-type and *mksr-1* worms we observe that the initial rate of recovery after photobleaching is significantly higher in mutant worms (Supplementary Fig. 7a–c). This suggests that in *mksr-1* mutants IFT components enter faster, either because of the higher IFT velocity, different IFT train sizes, and/ or because of the cilium being a more open system. Finally, the super-resolution map of OSM-3 in *mksr-1* worms indicates that the cilium maintains its hollow structure in the proximal segment, bulging more than in wild type (Supplementary Fig. 8a, b). Furthermore, to our surprise, the disruption of the ciliary structure of *kap-1* (which does not bulge in the PS) appears partially recovered in *kap-1*; *mksr-1* (showing modest bulging; Supplementary Fig. 8a, b). Thus, we find that in TZ mutants, kinesin-2 motors enter the cilium more readily, driving anterograde motion significantly faster, while the axoneme maintains its structural integrity.

## Discussion

In this study, we imaged single IFT proteins at the ciliary base of the PHA and PHB chemosensory neurons in *C. elegans*, using SWIM[23]. We observed that anterograde IFT motors, kinesin-II and OSM-3, as well as cargoes of anterograde IFT trains, IFT dynein and tubulin, are present in a diffusive pool at the PCMC, on the dendritic side of the ciliary base.

From this pool, IFT proteins are recruited to the ciliary base by a diffusion-to-capture mechanism, as hypothesized before[13] and are stuck there for a while ("dock"), before starting their anterograde journey through the cilium. We observed remarkable differences between the different IFT proteins. IFT dynein docks in a narrow region of space at the ciliary base and remains stuck for on average in the range 2.7–8 s before being transported into the cilium (Fig. 3f). Kinesin-II motors remain docked for only a short while, 0.4–0.8 s (Fig. 2h), while OSM-3 and tubulin do not pause at all but start their anterograde journey immediately after docking. This points towards a sequential mechanism of anterograde IFT-train formation (Fig. 7) where the core of the IFT trains is first formed by IFT-particle complexes, to which IFT dynein binds, followed by kinesin-II. Tubulin and OSM-3 only bind to trains that are already moving. Our findings in *C. elegans* are aligned with recent studies in *C. reinhardtii*. A FRAP study showed that IFT trains take on average ~9 s to assemble at the ciliary base[12] and a structural study revealed that the ciliary base is lined with anterograde IFT trains in different stages of assembly, with IFT-A and IFT dynein linearly oligomerizing from front to back on an IFT-B scaffold[14]. These studies and ours indicate that kinesin-2 motors only associate with IFT trains in a late stage of their assembly. Binding of kinesin-2 to IFT trains releases its autoinhibition and allows the motors to engage with the axoneme, driving the train forward. Also tubulin, which associates as cargo with IFT-B complexes in moving anterograde trains[20], does not appear to bind to the IFT trains during assembly. Subtle structural differences have been observed between assembling and moving trains, suggesting that assembling trains undergo conformational changes in the last stages of assembly[14]. Further experiments will be required to unravel whether it is this conformational change that allows only some IFT components to dock to assembling trains.

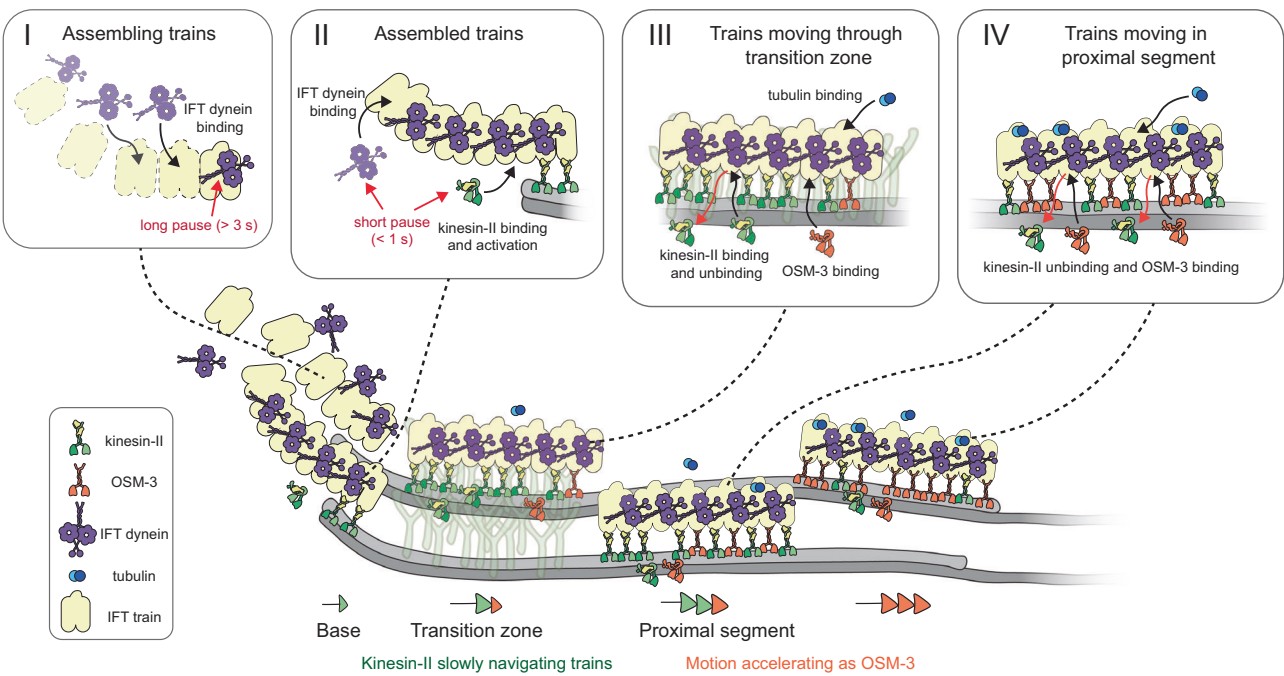

**Fig. 7 | Illustration of entry dynamics and movement of IFT motors and tubulin in cilia of *C. elegans*.** Kinesin-II, OSM-3, IFT-dynein and tubulin associate with anterograde trains at different stages. IFT-dynein molecules dock onto assembling IFT trains (see box I). When it binds to an anterograde train during the early stage of assembly, it pauses for a long duration (several seconds). Once the anterograde IFT train is assembled, diffusing kinesin-II molecules associate with the trains, switching into an activated state (see box II). Once several kinesin-II molecules attach, the train slowly moves forward. The last IFT-dynein molecules can still associate with trains in the latest stages of assembly, pausing for a short duration (<1 s). Trains move through the TZ slowly, primarily driven by kinesin-II motors which are well suited for navigation through the dense environment (see box III). Motor (dis) association with the trains is dynamic, with kinesin-II motors rapidly detaching (and reattaching) and both kinesin-II and OSM-3 attaching to the trains. At this stage, tubulin can also associate with the moving trains, as cargo. In the proximal segment, unhindered trains are swiftly driven by kinesin-II and OSM-3 motors (see box IV). The motion of the trains accelerates while they move further into the cilia, since OSM-3 gradually replaces dissociating kinesin-II motors. Diffusing tubulin molecule can dock onto such fast moving trains as cargo. Illustration adapted from Mul et al.[3].

We showed that IFT dynein only enters cilia as cargo of anterograde IFT trains (Fig. 3e). Tubulin, however, associates with anterograde trains throughout the TZ as well as in the PS (Fig. 3g, h), suggesting that it can diffuse through the TZ. This observation is consistent with studies in *C. reinhardtii*, where tubulin has been reported to diffusively enter cilia[21,22]. It is well established that the TZ forms a physical boundary preventing membrane proteins and large soluble proteins to freely diffuse into cilia[29–31]. In line with this, we observed that relatively small proteins like tubulin, and to a lesser extent kinesin-2, can freely diffuse over the TZ, while larger protein complexes like IFT dynein and IFT trains are blocked and require IFT to cross the TZ. In *mksr-1* mutant worms, where the organization of the TZ is compromised[8], we observed that kinesin-2 motors enter more readily and faster (Fig. 6 and Supplementary Fig. 7), with OSM-3 associating with trains deeper into the cilia (Fig. 6a, b). It would be interesting to further explore the role of TZ in entry of IFT proteins by studying the dynamics of IFT components in mutants, like *cep-290* or *mks-5* mutants[32], where the TZ structure has been shown to be more severely disrupted than in *mksr-1* worms.

From the single-molecule trajectories we calculated the velocities of the different IFT components in the first 2-3 μm of the cilium (Fig. 3i), providing important insights into IFT in the TZ. First, we found that the velocities of all IFT components studied are significantly lower in the densely crowded TZ than further in the cilium, as has been observed before[8,27]. Furthermore, in *mksr-1* cilia, we observed that kinesin-II and OSM-3 move faster in the TZ and initial part of the PS, in comparison to wild type. These observations agree with previous studies showing that disrupting the TZ structure results in IFT trains moving faster across the TZ[8,33]. Next, we found that the velocity of IFT components gradually increases in the first ~0.6 μm of the TZ, after which the velocity remains relatively constant (Fig. 3i). After ~1 μm, where the TZ ends, the velocity appears to increase again. Anterograde trains have a length of on average ~300 nm (in *C. reinhardtii*), and consist of many IFT-B complexes, repeating every 6 nm, that provide binding sites for the kinesin-2 motors[5,6]. Most likely, when an IFT train takes off from the ciliary base on its anterograde journey, only a few kinesin-2 motors in the front of the train pull it forward along the axoneme. After entering the TZ, more and more motors bind to the train and contribute to driving its anterograde transport. We hypothesize that after ~0.6 μm into the TZ, anterograde trains have a saturating number of kinesin motors engaged with the axoneme lattice, such that trains have acquired a constant velocity in the crowded TZ. As trains move further than ~1 μm into the cilium, they start moving out of the TZ, resulting in less restriction by the crowded environment and further acceleration.

We also studied anterograde IFT in mutant worms lacking kinesin-II function. In these worms OSM-3-driven anterograde IFT accelerated faster beyond the TZ, with the maximum OSM-3 velocity reached earlier in the cilium than in wild type (Fig. 5d). This has been observed before and is due to the faster OSM-3 being the only motor responsible for anterograde transport[2,3]. In wild-type worms, the region immediately beyond the TZ coincides with the "handover zone" where the slower kinesin-II starts falling off trains with more of the faster OSM-3 binding, resulting in a more gradual acceleration. Also, we observed that anterograde IFT velocity in *kap-1* worms is substantially lower in the TZ, which indicates that, while OSM-3 takes over the role of kinesin-II as importer of IFT trains and transporter in the TZ, it is less efficient at it. Indeed, in vitro studies have suggested that the slower kinesin-II is much better at navigating crowded environments than the faster OSM-3[34,35].

Another striking observation was that the velocity of kinesin-II, and to a lesser extent OSM-3, is lower than those of the cargoes IFT-dynein and tubulin (~20–30%; Fig. 3i). Faster single-molecule imaging of IFT motors revealed that this is likely caused by differences in kinetics of binding to and unbinding from anterograde trains. We observed that kinesin-II rapidly docks on and off anterograde trains in the TZ, switching into short diffusive states[25], not detectable with the time resolution of our experiments. This would result in a lower average velocity of single kinesin-II and OSM-3 motors. Also, rapid unbinding from IFT trains and rebinding after a diffusive episode could be an additional strategy to navigate in a dense environment. Finally, in contrast to both the kinesin-2 motors, it is possible that IFT dynein and tubulin, both cargoes of anterograde IFT trains, remain tightly coupled to the anterograde trains while they move across the TZ, resulting in the measured velocity being significantly faster.

The single-molecule trajectories of IFT components allowed generation of super-resolution maps, shining light on the functional role of kinesin-II in *C. elegans* sensory cilia. A remarkable observation was that the ciliary axoneme appears to "bulge" out beyond the TZ in the PS of wild-type cilia. This bulge is absent in *kap-1*, as has been reported before[27]. Furthermore, in these mutants the structure of the ciliary base is altered in comparison to wild type (Fig. 5g–i). We also observed that IFT dynein associates with anterograde trains not only at the base, but also in the TZ (Fig. 5f). In addition, the distribution of IFT components in the PS appears less hollow in the center than in wild-type worms, which indicates that the 9 microtubule doublets are more irregularly distributed and do not form a symmetric hollow cylinder (Fig. 5g). It might be that this hollow cylindrical architecture of the axoneme is necessary to properly orient trains such that they can interact with both the axoneme and the ciliary membrane. Taken together, these observations demonstrate that, although OSM-3 can build the cilium on its own, ciliary ultrastructure at the base, TZ and PS appear compromised in *kap-1* mutant worms.

Finally, we observed that pause durations of IFT-dynein were significantly shorter in *kap-1* mutant cilia (in the range 1.5–3.5 s in comparison to 2.7–8 s in wild type). Before, we have measured that both frequency and size of anterograde IFT trains are lower in *kap-1* mutant cilia than in wild type[8]. Taken together, these observations indicate that, while OSM-3 is capable of taking over anterograde IFT through the TZ in *kap-1* mutants, IFT-train assembly is substantially hampered: train assembly takes shorter duration and trains depart prematurely. This might indicate that kinesin-II plays a role in regulating IFT-train assembly and departure at the ciliary base. Further experiments will be needed to unravel the molecular basis of kinesin-II in these processes.

In summary, we have unraveled the dynamics of individual IFT motors and cargo proteins entering the phasmid chemosensory cilia of *C. elegans*, using quantitative single-molecule imaging. Such single-molecule imaging is also possible for other cilia in worms, although factors including architectural complexity, more frequent twitching of the worm body and higher background autofluorescence provide a challenge for quantitative single-molecule analysis. Nonetheless, a qualitative single-molecule view of IFT-dynein in an amphid cilia bundle (Supplementary Movie 6) suggests that the mechanism of association of IFT motors to anterograde IFT trains may be similar for other *C. elegans* cilia. Looking ahead, our approach can be extended to visualize the entry and exit of diverse IFT components, leading to a comprehensive understanding of the mechanisms responsible for maintaining and regulating the heterogeneous pools of proteins within the cilium.

## Methods

### C. elegans strains
The worm strains used in this study are listed in Supplementary Table 1. The strains used have been generated before in our laboratory, using Mos-1 mediated single-copy insertion[36]. Maintenance was performed using standard *C. elegans* techniques[37], on NGM plates, seeded with HB101 *E. coli*.

### Fluorescence microscopy
Images were acquired using a custom-built laser-illuminated widefield fluorescence microscope, as described previously[38]. Briefly, optical imaging was performed using an inverted microscope body (Nikon Ti E) with a 100x oil immersion objective (Nikon, CFI Apo TIRF 100x, N.A.: 1.49) in combination with an EMCDD camera (Andor, iXon 897) controlled using MicroManager software (v1.4). 491 nm and 561 nm DPSS lasers (Cobolt Calypso and Cobolt Jive, 50 mW) were used for laser illumination. Laser power was adjusted using an acousto-optic tuneable filter (AOTF, AA Optoelectronics). For performing small-window illumination microscopy (SWIM)[23], the beam diameter was changed using an iris diaphragm (Thorlabs, SM1D12, ø 0.8–12 mm) mounted between the rotating diffuser and the epi lens, at a distance equal to the focal length of the latter. The full beam width in the sample was ~30 μm (2σ of the Gaussian width). The aperture size of the diaphragm was adjusted manually to change the width of the beam, with a minimum beam width of ~7 μm at the sample, when the diaphragm is closed to a minimum diameter of 0.8 mm. Fluorescence light was separated from the excitation light using a dichroic mirror (ZT 405/488/561; Chroma) and emission filters (525/50 and 630/92 for collecting fluorescence excited by 491 nm and 561 nm, respectively; Chroma).

### Image acquisition
**Sample preparation.** For imaging live *C. elegans*, young adult hermaphrodite worms were sedated in 5 mM levamisole in M9, sandwiched between an agarose pad (2% agarose in M9) and a coverslip, and mounted on a microscope[38].

**Single-molecule imaging using SWIM.** To perform SWIM, a small excitation window (width ranging between 10–15 μm) was used to illuminate fluorescent molecules (IFT proteins labeled with eGFP or mCherry) in a pair of PHA/PHB cilia, as well as a small section of the corresponding dendrites (see example movies in Supplementary Movies 1–5). For single-molecule imaging, a high intensity 491 nm or 561 nm beam (~10 mW/mm² in the center of the beam) was used to illuminate the sample. Only those worms were imaged where both the proximal and middle segment (at least ~70% of the ciliary length) of one or both cilia from the cilia pair were uniformly in focus in the focal plane, with the distal region often slightly out of focus. Upon sustained high-intensity laser excitation of the cilium pair, a single-molecule regime is reached when all the labeled protein inside the cilia are photobleached and only proteins freshly entering the small excitation window are detected with fluorescence. This single-molecule regime was reached within ~1 min for mCherry-labeled OSM-3 and ~2–5 min for eGFP-labeled kinesin-II and IFT-dynein, depending on the exact location of the excitation window with respect to the ciliary base. For tubulin, reaching the single-molecule regime took longer (>5 min) since tubulin is incorporated within the axoneme lattice, which takes longer to completely photobleach. Image acquisition for analyzed data was performed in the range 6.6–20 fps, with most data acquired at 6.6 fps. Samples were typically imaged for 10–45 min. For the example movies in Supplementary Movie 3, imaging was performed at 31 fps for OSM-3 and 60 fps for kinesin-II and IFT-dynein.

**FRAP acquisition.** To image fluorescence recovery of OSM-3::mCherry after photobleaching, in wild-type and *mksr-1* mutant cilia, the following sequence was used: first, the prebleached cilium pair was imaged every 5 s at 5% of the maximal 561 nm laser intensity for 12 frames, with exposure time per frame 150 ms. Then the cilium pair was photobleached over 2.5 min by continuous illumination with the

561 nm laser at maximal power, using a small excitation window to photobleach only the cilium pair. Finally, to image the fluorescence recovery, the photobleached cilium pair was imaged every 5 s at 5% of the maximal 561 nm laser intensity for 15 min, with exposure time per frame 150 ms.

## Image analysis

Single-molecule tracking was only performed on those worms that did not show micron-range movement during the entire acquisition time (ranging between 10–45 min). Movement of the worm or image drift in the nanometer range could not be accounted for and could have a minor impact on the numbers we acquire from our analysis. To account for this one would require having fiducial markers, that do not bleach, within the body of the worm being imaged, which is technically challenging. The localization precision of individual fluorescing molecules is likely in the range of 40 nm ($2\sigma$), as estimated for surface-bound eGFP in our experimental set-up[8], although it is slightly higher for moving molecules due to motion blur[39].

**Tracking.** Single-molecule events were tracked using a MATLAB-based software, FIESTA (version 1.6.0)[40]. Tracks, corresponding to an event, contain information regarding time ($t_i$), x and y coordinates ($x_i$, $y_i$) and distance moved ($d_i$), for every frame $i$. The connected tracks were visualized and only tracks, corresponding to single-molecule entry events starting at (or near) the ciliary base, were included for further analysis, with a minimum track length of 12 frames. All the observed IFT proteins (kinesin-II, OSM-3, IFT dynein and tubulin) diffuse in the dendrite and switch to a paused or directed state on associating with an IFT train. This switch results in a clear increase in localized intensity, marking the starting frame of the track. If diffusion was observed in a few frames before association with the IFT train, these frames were removed from the track. Erroneous tracks, primarily caused by two single-molecule events too close to discriminate, were also excluded from further analysis (though used for fitting the spline). It is possible, however, that occasionally molecules tracked as single molecules were due to more than one fluorescing molecule binding to the same train within a short time interval. It is difficult to discern this at our acquisition rate (6.6 fps), but we deem it negligibly rare.

**Transformation to ciliary coordinates.** A ciliary coordinate system was defined by interpolating a cubic spline on a segmented line drawn along the long axis of the imaged cilium, visualized via the single-molecule localizations obtained from tracking (see Supplementary Fig. 2). A reference point was picked at the base of the characteristic "bone-shaped" structure, at the ciliary base. All single-molecule localizations were transformed from x and y coordinates ($x_i$, $y_i$) to ciliary coordinates ($c_{\parallel\_i}$, $c_{\perp\_i}$), with $c_{\parallel}$ the distance from the reference point along the spline and $c_{\perp}$ the distance perpendicular to the spline ($c_{\perp}$), as illustrated in Fig. 2a.

**Velocity measurement.** Before calculating the point-to-point velocity, the tracks were smoothened by rolling frame averaging over 10 consecutive time frames, to reduce the contribution due to localization error (typically estimated to be between 10-80 nm, depending on the brightness of the tracked object). The point-to-point velocity at a given localization ($x_i$, $y_i$) was calculated using the following equation:

$$v_i = (d_{i+1} - d_{i-1})/(t_{i+1} - t_{i-1}) \tag{1}$$

Only moving data points (see details of classification method below) are displayed in the Figures, where we calculated the average velocity and error of the velocity data at bins of 100 nm along the cilia's long axis, using bootstrapping.

**Docking location.** Docking location of a single-molecule track was defined as the averaged coordinates ($c_{\parallel}$, $c_{\perp}$) of the first 2 frames.

**Measured pause time.** Measured pause time, $t_{p\_m}$, of a single-molecule track was defined as the time taken (interpolated) to move the first 100 nm. Tracks shorter than 300 nm were discarded from the distribution.

**Bleach time fit.** The characteristic bleach time $t_{bleach}$ for a given fluorescently labeled protein in the cilia was obtained from the exponential fit ($e^{-x/t_{bleach}}$) to the decay in the fluorescence intensity (normalized) over time, as a result of bleaching induced by the high intensity laser exposure (as shown for KAP-1::eGFP in Supplementary Fig. 3b). The fluorescence intensity was measured on ImageJ by manually selecting a region containing fluorescent signal and a region next to it within the illuminated area, for background correction.

**Numerical simulations of single-molecule trajectories.** To determine the impact of bleaching on the measured pause time, entry events were numerically simulated (scheme illustrated in Supplementary Fig. 3a). Simulated tracks were allowed to dock at a given location along a 1D cilia lattice, with the docking location randomly picked from the distribution of docking locations obtained experimentally. Each track was assigned a bleach time ($t_b$) and a "real" pause time ($t_p$) randomly picked from exponential distributions with characteristic time $t_{bleach}$ (obtained from experiments) and $t_{p\_actual}$ (free parameter), respectively. The location of the simulated event was updated every $\Delta t$ (with $\Delta t = 150$ ms, as in most experiments). The localization precision $\Delta$ was estimated to be 40 nm ($2\sigma$). At a frame $i$, while $t_i < t_b$ or $D_i < 6$ μm (2 μm in case of kinesin-II simulation), the location of the event was updated as follows:

$$\begin{aligned} D_1(0) &= \Delta_1; \; if \, i = 1 \\ D_i(t_i) &= D_{i-1}(t_i - \Delta t) + \Delta_i; \; if \, t_i < t_p \, \text{or} \\ D_i(t_i) &= D_{i-1}(t_i - \Delta t) + v_i \times \Delta t + \Delta_i; \; if \, t_i > t_p \end{aligned} \tag{2}$$

where, $v_i = v_{avg\_D_{i-1}} + \Delta v_{std\_D_{i-1}}$ ($v_{avg}$ is the location-dependent velocity at $D_{i-1}$ obtained from experiments; $\Delta v_{std}$ is a randomly picked value from a normal distribution with width being the error in velocity at $D_{i-1}$ obtained from experiments) and $\Delta_i$ is the localization precision, randomly picked from a normal distribution centred at 0 with width 40 nm. For each simulation experiment, with number of events $N$, the "actual" pause time, $t_{p\_actual}$, is varied and the measured pause time, $t_{p\_m}$, is recorded for tracks longer than 300 nm, as described for experimental data (see Fig. 2e). For simulations corresponding to every IFT component, we scan $t_{p\_actual}$ by steps of 0.01 and obtain the corresponding average $t_{p\_m}$ (and error). $t_{p\_m}$ values obtained from our simulations that match the experimentally measured $t_{p\_m}$ provides an estimate of the actual pause time for a given IFT component. For most of the simulations N = 1500, except for simulations corresponding to IFT dynein ($N = 10,000$), where a much smaller fraction of simulated events are longer than 300 nm in the experimentally relevant range of pause times.

The underlying distribution of measured pause time is estimated to be a convolution of two exponentials with rates $k_{p\_actual}$ ($1/t_{p\_actual}$) and $k_{bleach}$ ($1/t_{bleach}$) that can be defined by the following equation:

$$y(t) = \frac{(k_{pactual} \times k_{bleach})}{(k_{pactual} - k_{bleach})} \times (e^{-k_{bleach}t} - e^{-k_{pactual}t}) \tag{3}$$

We chose to perform numerical simulations instead of fitting the data analytically since the measured pause time is different from the pause time the analytical solution provides (which cannot be measured experimentally).

**Classification of directed transport and pausing.** To obtain a quantitative measure for the directedness of the motion, we used an MSD-based approach to extract the anomalous exponent ($\alpha$) from $MSD(\tau) = 2\Gamma\tau^{\alpha}$ (where $\Gamma$ is the generalized transport coefficient and $\tau$ is the time lag) along the track, in the direction of motion. $\alpha$ is a measure of the directedness of the motion, $\alpha = 2$ for purely directed motion, $\alpha = 1$ for purely diffusive motion and $\alpha < 1$ for sub-diffusion or pausing. For each datapoint ($c_{\parallel\_i}$), we calculated $\alpha$ in the direction parallel to the spline ($\alpha_{\parallel\_i}$), using a windowed Mean Square Displacement classifier (wMSDc) approach, described in Danné et al.[41]. $\alpha$ was calculated analytically, using the following equation:

$$\alpha = \langle\alpha\rangle = \frac{2}{W-1}\sum_{n=1}^{(W-1)/2} \log\left(\frac{MSD\,(n\Delta t + \Delta t)}{MSD\,(n\Delta t)}\right) / \log\left(\frac{n+1}{n}\right); \quad (4)$$

keeping a fixed window ($W = 12$ frames). Due to the size of the window, all tracks shorter than 12 frames were removed from the analysis. Datapoints with $\alpha_{\parallel\_i} > 1.2$ are classified as directed and $\alpha_{\parallel\_i} < 1$ are classified as static.

**Numerical simulations to generate 1D projection library.** Distribution of 2D localizations ($y_i$, $z_i$) along a transverse cross-section of the modeled hollow cylinder (scheme illustrated in Fig. 4c), was numerically simulated ($N = 10,000$), with $r_{ax}$ providing the width of the cylinder, $r_{db}$ providing the thickness of the ring and localization precision estimated to be 40 nm ($2\sigma$ for normal distribution). Coordinates of individual localizations were obtained as follows:

$$y_i = r_{ax} \times \cos\theta_{ax} + r_{db} \times \cos\theta_{db} + \Delta_i$$
$$z_i = z_{ax} \times \sin\theta_{ax} + r_{db} \times \sin\theta_{db} + \Delta_i \quad (5)$$

where, $\theta_{ax}$ and $\theta_{db}$ was randomly assigned to the localization and $\Delta_i$ (localization precision) was randomly picked from a normal distribution centred at 0 with width 40 nm. The distribution of $y_i$, a bimodal distribution centered at 0, was fitted with a kernel density estimation which provided the 1D projection distribution of localizations along the transverse cross-section of a hollow cylinder. A library of 1D projection distributions was generated for varying $r_{ax}$ (ranging from 100–300 nm with steps of 5 nm) and $r_{db}$ (ranging from 5–50 nm with steps of 1 nm). For one condition ($r_{ax} = 150$ nm and $r_{db} = 25$ nm) 1D projection distributions were obtained for varying $\Delta_i$ (ranging from 10-50 nm with steps of 10 nm; Supplementary Fig. 5g). In one case ($r_{ax} = 150$ nm, $r_{db} = 25$ nm and $\Delta_i = 40$ nm; Supplementary Fig. 5h, i), a Gaussian filter of width 300 nm ($2\sigma$ for normal distribution) was applied along the z-axis, in order to undersample the localizations above and below the center of the distribution.

**Estimating the underlying 3D distribution from IFT-dynein single-molecule localizations.** The absolute distance perpendicular to the cilia spline was sampled every 1000 data points along the long axis of the cilia (shifting every 300 data points for the next sample), from the single-molecule localization map of IFT-dynein (Supplementary Fig. 5a, b). This distribution was mirrored around zero and a kernel density estimation of the distribution was obtained (Supplementary Fig. 5c). By using maximum-likelihood estimation (MLE), the most accurate estimate of this distribution in the 1D projection library was determined, which allowed us to estimate the $r_{ax}$ and $r_{db}$.

**Estimating the shape of the cilia from single-molecule localizations.** The absolute distance perpendicular to the cilia spline was sampled every 1000 data points along the long axis of the cilia (shifting every 300 nm for the next sample) from the single-molecule localization map of a given imaged species. The 80-percentile value obtained from the cumulative distribution function of the sampled distribution provided the width of the cilia at the location along the cilia spline where

the distribution was sampled. We chose the 80-percentile value of the cumulative distribution function since this is approximately where the distribution of absolute distance perpendicular to the cilia spline peaks for sampled IFT-dynein localizations. In total, 70–90 percentile range of the cumulative distribution function was displayed to represent the error in the width.

**Estimating average value and error for distributions.** We used a bootstrapping method to calculate the parameters of a distribution. We randomly selected N measurements from the distribution (with replacement) and calculated the median of the resampled group. We repeated this process 1000 times, creating a bootstrapping distribution of medians. We then calculated the mean ($\mu$) and standard deviation ($\sigma$) of this distribution, and used these values to estimate the parameter and its error. In this paper, all values and errors are presented as $\mu \pm 3\sigma$.

**Estimating fluorescence recovery rate and recovery fraction from FRAP acquisition.** The fluorescence intensity of each cilium pair was measured using ImageJ by manually selecting a region containing the fluorescent signal from a given cilium pair and a region next to it within the illuminated area, for background correction. The intensity was normalized to the average prebleached intensity. By fitting the normalized intensity over time with the equation:

$$I_{norm}(t) = A \times (1 - e^{-b \times t}), \quad (6)$$

we obtained the fluorescence recovery fraction A. The initial rate of fluorescence recovery was obtained from the slope of a linear fit to the normalized intensity over the first 10 timeframes (2 min) after photobleaching.

**Information on plots and figures.** Kymographs were generated either on FIESTA or using the KymographClear[42] plug-in on Fiji/ImageJ. All the data was analyzed and plotted using custom written scripts on MATLAB (The Math Works, Inc., R2021a).

**Reporting summary**
Further information on research design is available in the Nature Portfolio Reporting Summary linked to this article.

## Data availability
The underlying experimental data in this study can be found in the Source data file. Further, the same data along with scripts (written on MATLAB) to visualize the data as displayed in the figures are available on DataverseNL [https://doi.org/10.34894/EBPWXD] Source data are provided with this paper.

## Code availability
MATLAB scripts for data analysis, visualization and numerical simulations associated with this study are available on DataverseNL [https://doi.org/10.34894/EBPWXD].

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

## Acknowledgements

We thank Dr. Misha Klein for discussions regarding the pause-time simulations, Dr. Bram Prevo for generating worm strain EJP72 and members of our lab for fruitful discussions. We acknowledge financial support from the European Research Council under the European Union's Horizon 2020 research and innovation programme (Grant agreement no. 788363; "HITSCIL"; E.J.G.P.) and Marie Sklodowska-Curie Actions Postdoctoral Fellowship of the European Commission (Project no. 898006; 'MingleIFT', A.M.).

## Author contributions

Conceptualization and methodology: A.M. and E.J.G.P.; investigation: A.M. and E.L.; formal analysis: A.M.; resources: A.M.; visualization: A.M. and E.L., writing—original draft: A.M.; writing—review & editing: A.M., E.L. and E.J.G.P.; funding acquisition: A.M. and E.J.G.P.; supervision: E.J.G.P.

## Competing interests

The authors declare no competing interests.
