## [Peer Review File · Nature Communications]

IFT cargo and motors associate sequentially with IFT trains to enter cilia of *C. elegans*REVIEWER COMMENTS

Reviewer #1 (Remarks to the Author):

This manuscript uses single-molecule fluorescence imaging to investigate the dynamic assembly of anterograde IFT trains at the ciliary base in *C. elegans* sensory neurons. While the field mostly focused on how the cargos reach the ciliary tip and disassemble, until recently, not much was known about how the trains assemble at the base. Recent cryoelectron tomography (cryo-ET) showed the ultrastructure of the IFT trains that assemble at the base of *Chlamydomonas* flagella and showed that IFT-A and IFT-B trains assemble first, which then recruit dynein motors, whereas kinesin-2 is recruited to these trains near the transition zone (TZ). In this study, the authors have tracked the entry of IFT dynein, kinesin-2, OSM-3, and tubulin in *C. elegans* cilia and determined where they stop at the base and how long it takes for these proteins before they leave TZ and move along the cilia. They showed that IFT dynein docks onto IFT trains for about 9 s at the ciliary base. Kinesin-2 remains docked for a shorter period of time, whereas OSM-3 and tubulin do not pause at the base and can hop onto moving anterograde trains before they exit TZ. These results are largely consistent with the cryo-ET study in *Chlamydomonas* and provide information about the dynamics of IFT assembly.

While this manuscript potentially has several merits, I have several major concerns that need to be addressed before I can recommend publication in Nature Communications.

1. It is unclear why OSM-3 and tubulin do not attach to the paused trains but to the moving trains. Is it possible that the distinction between diffusive entry into the cilia and the start of the unidirectional motion cannot always be reliably made at such a short distance (0-400 nm) especially given that these proteins do not exhibit a clear pause before they move? What is the error in determining the docking position, especially in the absence of long pausing behavior? Because there is still a significant overlap in the docking positions of dynein and OSM-3, it is possible that OSM-3 can also bind to paused trains at the base while the others can hop on the trains that just started moving/ diffusing between the base and TZ.
2. Do the IFT trains exhibit clear unidirectional movement or diffusion as they move beyond the docking position of dynein (between 400-1600 nm in Figure 2)? This should be shown more clearly in a main text figure.
3. "Velocity" measurements in the Results section are quite misleading, as IFT particles, dynein, kinesin-2, Osm-3, and tubulin all move together unidirectionally and cannot have different velocities from each other. In the Discussion, the authors clarify why this could be the case (i.e. rapid attachment/detachment of motors from IFT), but the authors need to either clarify this point early in the Results section or replace "velocity" with another definition (such as time it takes to traverse between the base and TZ) to explain their measurements.
4. The authors use single-molecule trajectories to estimate the space at which the cargos and motors can move at the ciliary base. While everything looks normal in native conditions, and in OSM3 mutants, trajectories in the kinesin-2 mutants are more disorganized. Based on this observation, the authors claim

that the structure of the ciliary base is disrupted. They wrote, “although OSM-3 can build the cilium on its own, ciliary ultrastructure at the base, TZ and appear substantially compromised in kap-1 mutant worms.” Based on these very limited observations, this statement is too strong and probably incorrect, given that the observed effect is not seen in the kap1/mksr-1 double mutant. The authors need to directly test their conclusion by determining the ciliary ultrastructure using well-established methods in cryo-ET.

Reviewer #2 (Remarks to the Author):

Review comments

In this work, Mitra and colleagues use a single molecule imaging approach to investigate the behaviour of fluorescent-protein tagged IFT motors (kinesin-II, OSM-3, IFT-dynein) and cargo (tubulin) within the proximal-most regions (eg. 1-2 microns encompassing the basal body and transition zone) of sensory cilia in *C. elegans*. Using small-window illumination microscopy (SWIM) in living worms, the authors generated time-lapse recordings of reporter localisations and movements, from which they extracted various types of data (eg, reporter docking sites, pause times, speeds) in WT animals and worms with disrupted IFT motors and transition zones. These data were then used to draw conclusions concerning the spatial and temporal aspects of anterograde IFT train assembly at the ciliary base, and the role of two kinesin 2 motors in this process.

The authors report different ciliary base pause behaviours for the different reporters, with OSM-3 and IFT-dynein pausing for shorter and longer periods, respectively, compared with kinesin-II. Such data is consistent with the motors associating with IFT trains at different stages of their assembly (ie. IFT dynein associates earlier than kinesin-II; OSM-3 associates with trains that are almost fully assembled or fully assembled and starting to move through the TZ). Tubulin hardly pauses at the TZ indicating that it is picked up by fully assembled and moving trains. In terms of docking site, the SWIM imaging shows that whilst kinesin-II and IFT-dynein dock at the proximal part of the TZ, OSM-3 docks over a wider area (and tubulin wider still), consistent with the model of kinesin-II being the initial primary driver of anterograde IFT, after which OSM-3 takes over (as shown previously by this group). For the analyses in IFT motor mutants, the paper shows that kinesin-II docking location, pause time and velocity at the ciliary base are not affected by OSM-3 loss, whereas the reverse is not the case. Specifically, in worms lacking kinesin-II function, OSM-3's pause times increase (at least 1.8 times), its docking location is broader (includes the basal body region), and its initial velocity reduces. IFT-dynein also docks to a wider region of the ciliary base, and shows reduced initial velocities, in kinesin-II mutants. From these data, authors conclude that in the absence of kinesin-II, OSM-3 can take over anterograde IFT transport at the ciliary base, albeit in a less efficient manner. The paper finishes by showing that in worms lacking the TZ regulator mskr-1, anterograde train initial velocities are increased and that OSM-3 can dock over a wide region of the TZ;

authors conclude that the TZ acts to impede kinesin-II entry into cilia and the initial velocity of anterograde IFT trains.

This paper provides some important insight into the spatial and temporal association of IFT motors and cargo (tubulin) with anterograde IFT trains as they assemble at the ciliary base, and subsequently move through the TZ into the axoneme proper. The single molecule imaging data is of high quality and generally supports the conclusions being made, though not always in all cases (see below).

Major comments

1. Although the paper is well written, the figure layout of the data is not always optimal for following the results as they are presented in the main text. Mostly, the issue was that some result statements required the reader to consider data across multiple figures. For example, in Figure 4 (IFT motor mutants), you have to refer to Figs 1 and 2 to see most of the control (WT) data. Another example is for the results statement: “ the super-resolution maps of static OSM-3 and IFT dynein show

that the organization at the base is less well defined (Fig. 4G and Extended Data Fig. 6I) compared to

wild type (Fig. 3A)....”. In this statement, you have to look at the same type of data in 3 figures ! There are other examples like this in the paper. A figure should always include the relevant controls for ease of comparison. I can see how this is not so easy to achieve when multiple readouts are being presented for each sample; nonetheless, the authors should make an effort to better include control data in the relevant figures.

2. Pg 4: “Docking locations show a wider distribution than in wild type, with docking occurring in the TZ as well as at the ciliary base, where OSM-3 takes over the role of absent kinesin-II (Fig. 4B).” To me, the difference between WT and mutant in 2B vs 4B is very slight, if anything at all. This is important to clarify for the conclusions.

3. Pg 5: “Second, the super-resolution maps of static OSM-3 and IFT dynein show that the organization at the base is less well defined (Fig. 4G and Extended Data Fig. 6I) compared to wild type (Fig. 3A), with static localizations distributed in a single transversal distribution (instead of

two distributions symmetrical around the centre).” In Fig. 4G (left panel), you can still make out 2 distributions although they seem closer together compared to 3A. Whether this can be described as ‘less well defined’ is debatable.

4. Pg 5: “The distribution of the docking locations of OSM-3 in *mksr-1* worms extends much deeper into the TZ and the PS (Fig. 5A,5B), suggesting that OSM-3 can more readily enter the cilium when the TZ is

compromised. OSM-3 (Fig. 5F)". The difference in OSM-3 docking location between WT (334 nm) and mksr-1 mutant (428 nm) is less than an average of 100 nm; also, the histograms in 3A and 5B don't show much difference; thus, the notion that docking extends "much deeper into the TZ" is clearly overstated in my opinion. I would have liked to have seen data for another TZ mutant; for example, the cep-290 or mks-5 mutant where TZ structure is more severely affected than the mksr-1 mutant.

5. Why did the authors not measure the flux rate of IFT particles moving into the axoneme from the TZ, given the observation of increased OSM-3 and kinesin-II initial velocities in the TZ of mksr-1 mutant? Higher TZ train speeds could lead to more IFT trains entering the cilium per second, or not (either way, this would be very interesting to know).

6. Figure 3 claims to "reveal the structure at the proximal part of cilia" and "mapping of ciliary ultrastructure", concluding "In summary, super-resolution maps provided by single-molecule localizations allow accurate determination of the cylindrical shape of the ciliary proximal part, governed by the axonemal MT doublets." Whilst this is true, the conclusion does not really yield a lot of new insight, given the many studies in different systems showing IFT trains moving along the periphery of the hollow ciliary axoneme, sandwiched between the ciliary MTs and membrane. Maybe I am missing the point but what is the purpose of providing an "accurate determination of the cylindrical shape of the ciliary proximal part" when this is all known already?

Minor comments

1. Introduction: "most ciliary proteins require hitching a ride on an anterograde trains to cross the TZ..." We don't know how true this is. Certainly some proteins do, but how global and widespread remains to be shown. I don't see this wider distribution when comparing 4B with 2B.

2. Pg 6: "First, we found that the velocities of all IFT components studied are significantly lower in the densely crowded TZ than further in the cilium, as has been observed before^{8,27}. Furthermore, in mksr-1 worms with disrupted TZ, we observed that kinesin-II and OSM-3 move faster in the TZ and initial part of the PS than in wildtype." That the TZ supports slower IFT speeds, regulated by TZ complexes, has been shown before in *C. elegans* in <https://pubmed.ncbi.nlm.nih.gov/26392567/>. This paper must be cited.

Reviewer #3 (Remarks to the Author):

Summary

The intraflagellar transport (IFT) system to ferry cargoes into the cilia which is carried out by multiple proteins including kinesin, OSM, dynein and tubulin, working in tandem across the packed transition zones. In nematodes, this involves slower navigation by kinesin-II in the anterograde direction across the transition zone and faster transport by OSM-3 to the apex of the cilia. Mitra et al in this study make use of a novel imaging tool – small-window illumination microscopy (SWIM) to increase the signal-to-noise ratio and to decrease the out of focus autofluorescence so as to carry out single molecule tracking of the IFT motors in live worms. They carried out their experiments in the PHA and PHB phasmid neurons of the worms. Their tracking experiments were combined with simulations and nanoscale localization experiments. In addition, the authors provided data on wild-type and kinesin-deficient worms, detailing their importance in the organisation, and directed transport of cargoes within the cilia.

Major Comments

- The authors apply the novel imaging tool – SWIM to obtain the movement of IFT associated proteins in vivo. The authors should confirm the validity of this narrow-window approach in at least a different neuron within the worms. Is its use limited to only the thin and transparent tip, or could it be expanded to image other cilia for example in the chemosensory neurons in the head of *C. elegans*?
- The tagging of the ciliary proteins with eGFP suggests that the SWIM method might be imaging like FRAP (that is, bulk protein movement imaging rather than individual protein tracking). With the use of eGFP, even in the narrow window imaging approach, the authors cannot rule out that bright spots are representative of individual proteins separated by the distance of the diffraction limit – about 250nm. Therefore, the authors should verify their data with either photoactivatable-GFP, photoconvertible mEOS3 or Halo tag to verify single protein data acquisition.
- The authors should confirm that the differences seen in docking locations, and pause times between kinesin-II, dynein, and OSM-3 is not due to differences in the levels of expressions of the proteins. If double copy numbers of kinesin-II were present, does this in anyway reduce its mobility?
- The authors should plot the area under the curve for the velocity plotted for the proteins (kinesin-II, dynein, tubulin, and OSM-3) and determine whether they are statistically different or not. This should be applied to other figures as well.
- How are the 3D-distributions of tubulin on the axoneme, altered in kinesin-deficient and SOM-3 worms?
- Do drug inhibitors like monastrol of the IFT proteins like kinesin, similarly alter the super-resolution map?

Minor Comments

- The authors should correct for the repetition of 'that' in the sentence: 'From the single-molecule tracks, we calculated that the average velocity and observed that it increases...'
- How did the authors account for micro- or nano-movement of the worm tail during data acquisition?

Point-to-point rebuttal of the reviewers' comments

We would like to thank the reviewers for their positive feedback and constructive comments on our work. We have addressed several of the points raised by the reviewers with additional experiments and altered the manuscript accordingly (with textual changes marked as track changes).

The key additions are as follows: (1) We have imaged at a faster rate (up to 10x faster) to better visualize the dynamics of individual proteins. The new data confirms that we observe single molecules and it reveals, in greater detail, the dynamics of IFT motors moving across the transition zone (TZ). (2) We have performed FRAP measurements of OSM-3 in wild-type and *mksr-1* mutant worm, which indicate that the rate of entry of the motor is higher in mutant cilia. (3) We have imaged IFT-dynein in amphid cilia to illustrate that our observations are not specific to the PHA/PHB cilia located in the tail of the worm but generally applicable to other cilia in *C. elegans* as well. (4) Finally, we have reorganized the figures and adjusted the text to optimize readability.

Below, we provide a point-by-point reply to all the comments raised.

Reviewer #1 (Remarks to the Author):

This manuscript uses single-molecule fluorescence imaging to investigate the dynamic assembly of anterograde IFT trains at the ciliary base in *C. elegans* sensory neurons. While the field mostly focused on how the cargos reach the ciliary tip and disassemble, until recently, not much was known about how the trains assemble at the base. Recent cryoelectron tomography (cryo-ET) showed the ultrastructure of the IFT trains that assemble at the base of *Chlamydomonas* flagella and showed that IFT-A and IFT-B trains assemble first, which then recruit dynein motors, whereas kinesin-2 is recruited to these trains near the transition zone (TZ). In this study, the authors have tracked the entry of IFT dynein, kinesin-2, OSM-3, and tubulin in *C. elegans* cilia and determined where they stop at the base and how long it takes for these proteins before they leave TZ and move along the cilia. They showed that IFT dynein docks onto IFT trains for about 9 s at the ciliary base. Kinesin-2 remains docked for a shorter period of time, whereas OSM-3 and tubulin do not pause at the base and can hop onto moving anterograde trains before they exit TZ. These results are largely consistent with the cryo-ET study in *Chlamydomonas* and provide information about the dynamics of IFT assembly.

While this manuscript potentially has several merits, I have several major concerns that need to be addressed before I can recommend publication in Nature Communications.

1. It is unclear why OSM-3 and tubulin do not attach to the paused trains but to the moving trains. Is it possible that the distinction between diffusive entry into the cilia and the start of the unidirectional motion cannot always be reliably made at such a short distance (0-400 nm) especially given that these proteins do not exhibit a clear pause before they move? What is the error in determining the docking position, especially in the absence of long pausing behavior?

The observation that OSM-3, tubulin and, to a lesser extent, kinesin-II rarely pause at the ciliary base is indeed a striking one and is one of our key findings. All the proteins that we imaged here (IFT motors and tubulin) are present in a diffusive pool in the dendrite and the PCMC, as can be observed in the example supplementary movies. The image acquisition rate we used was 6.6 frames/s (time per frame ~150 ms). Within this time frame a diffusive protein can move around over a substantially large volume of the PCMC, moving rapidly in and out of focus. As a consequence, these diffusing molecules appear as a blurry background. As soon as a molecule engages with a paused train or a slowly moving train in the TZ, its diffusive motion is limited and the protein appears as a clear punctum that can be localized and tracked. This is the moment that we start tracking an event (time zero). In this way we are confident that we detect the moment when a certain molecule switches from a diffusive to a pause or directed state with great accuracy. To further convince the reviewer, we have now imaged IFT-dynein, kinesin-II and OSM-3 with 5-10x faster acquisition rates. Using the faster acquisitions we can often see puncta of individual molecules diffusing in the PCMC (whenever they diffuse primarily in the plane of focus) before docking onto a paused or moving train. We have added **new Extended Data Fig. 4 and Supplementary Movie 3**, where we provide example IFT-dynein, kinesin-II and OSM-3 tracks imaged with this faster acquisition rate. It is evident from these examples that individual diffusive molecules dock at the ciliary base resulting in a significant increase in intensity upon docking.

The localization error of individual fluorescing molecules is in the range of 40 nm (2σ), as estimated for surface-bound eGFP in our experimental set-up (Prevo et al., Nature Cell Biology 2015). This error would be slightly higher for moving molecules due to motion blur. In order to estimate the error in localizing molecules engaging with a moving IFT train we consider an OSM-3 molecule when it binds to a train moving on average at the speed of ~200 nm/s (estimated from the average velocity plot for OSM-3 in Fig. 2I). This molecule will move ~30 nm in 150 ms (our frame acquisition time). Thus, there would be a minor increase in the localization error for molecules moving in a directed manner in comparison to molecules binding to stationary trains. We now clearly state the expected localization error in the methods section and indicate that motion blur would result it to be slightly higher.

Because there is still a significant overlap in the docking positions of dynein and OSM-3, it is possible that OSM-3 can also bind to paused trains at the base while the others can hop on the trains that just started moving/ diffusing between the base and TZ.

We only observe individual OSM-3 motors docking at the ciliary base and moving without a significant pause, which we interpret as binding of OSM-3 to moving IFT trains. In case OSM-3 molecules would bind to paused trains, we would also observe long pausing events in our acquired movies.

As the reviewer points out, there is a significant overlap in the docking positions of IFT-dynein and OSM-3. We think that this implies that there is a region at the ciliary base where stationary assembling IFT trains and assembled moving IFT trains co-exist. While IFT-dynein can bind to both paused and moving trains, OSM-3 only binds to moving trains.

2. Do the IFT trains exhibit clear unidirectional movement or diffusion as they move beyond the docking position of dynein (between 400-1600 nm in Figure 2)? This should be shown more clearly in a main text figure.

In this study we image the dynamics of anterograde IFT motors (kinesin-II and OSM-3) and anterograde IFT cargo (IFT-dynein and tubulin), but not the IFT train components directly. We assume that when individual IFT components are pausing or moving unidirectionally they are associated with IFT trains and represent the dynamics of the trains (as we have reported before in earlier work). In contrast, when they are not associated with IFT trains, they are diffusive.

All the IFT-components studied here are diffusive in the PCMC with the TZ forming a physical barrier that prevents free diffusive entry into the cilium. We find that diffusive IFT-dynein is blocked completely at the entry of the TZ and has to engage with IFT trains in order to enter the cilia. In contrast, from the distributions of docking locations we find evidence that kinesin-II, OSM-3 and tubulin can on occasions diffusively move further into the TZ before binding to an IFT train. We have explained this more clearly in the main text.

It must be noted that when imaging at 6.6 frames/s we were not able to capture the diffusive motion of a molecule before they dock at the ciliary base, and hence we had made our interpretations based on docking locations. However, now we have imaged at a much faster rate and can more clearly observe the diffusive motion of different IFT components before they bind to IFT trains (discussed above; **new Supplementary Movie 3**), which confirms our interpretation.

3. “Velocity” measurements in the Results section are quite misleading, as IFT particles, dynein, kinesin-2, Osm-3, and tubulin all move together unidirectionally and cannot have different velocities from each other. In the Discussion, the authors clarify why this could be the case (i.e. rapid attachment/detachment of motors from IFT), but the authors need to either clarify this point early in the Results section or replace “velocity” with another definition (such as time it takes to traverse between the base and TZ) to explain their measurements.

We indeed observe that the apparent average velocity of kinesin-II in the TZ is lower than that observed for IFT-dynein, tubulin and OSM-3. In the discussion of our original manuscript, we had reasoned that this could happen if kinesin-II motors rapidly associate and dissociate from IFT trains, resulting in short diffusive bursts (when not associated with trains) in between directed motion, which we cannot observe at the slow image acquisition rates (~6.6 frames/s). This argument was motivated by the findings of a previous paper from our lab (Zhang et al., PNAS 2020) where we indeed observed that kinesin-II is constantly dissociating and reassociating with anterograde as well as retrograde trains during its motion. In contrast to kinesin-II, we had argued that OSM-3, tubulin and IFT-dynein do not dissociate from anterograde IFT trains.

We have now imaged IFT components at a much faster rate; (i) 10x (~60 frames/s) for IFT-dynein and kinesin-II, and (ii) 5x (~31 frames/s) for OSM-3 (tagged with mCherry which is less bright than eGFP). This provides us a faster time-resolution view of individual IFT components moving across the TZ. We observe that once docked to IFT trains, OSM-3 and IFT-dynein move unidirectionally into the cilia once they start moving. In contrast, kinesin-II shows far more variable dynamics. Some events remain associated with IFT trains, moving unidirectionally, while other events display a more complex dynamics, switching to short diffusive burst in between directed motion or pausing. When we look at the events at 6.6 frames/s both types of events appear unidirectional. Since we have done all our analysis on movies acquired at 6.6 frames/s we also

include kinesin-II tracks in our analysis where the kinesin-II molecule switches on and off IFT trains. This results in the lower average velocity for kinesin-II in the TZ in comparison to the other components. We now show the new data in **Extended Data Fig. 4 and Supplementary Movie 3** and discuss this point carefully in the Results instead of only addressing it later in the discussion (see **new** section in the Results: “**Kinesin-II switches on and off IFT trains resulting in a lower apparent velocity**”)

4. The authors use single-molecule trajectories to estimate the space at which the cargos and motors can move at the ciliary base. While everything looks normal in native conditions, and in OSM3 mutants, trajectories in the kinesin-2 mutants are more disorganized. Based on this observation, the authors claim that the structure of the ciliary base is disrupted. They wrote, “although OSM-3 can build the cilium on its own, ciliary ultrastructure at the base, TZ and appear substantially compromised in kap-1 mutant worms.” Based on these very limited observations, this statement is too strong and probably incorrect, given that the observed effect is not seen in the kap1/mksr-1 double mutant. The authors need to directly test their conclusion by determining the ciliary ultrastructure using well-established methods in cryo-ET.

Our ciliary shape analysis reveals that wild-type cilia bulge after the TZ., Remarkably, this bulging is absent in cilia of worms lacking kinesin-2. This finding is robust given the consistency of this observation in large datasets acquired for multiple IFT components in both wild-type and mutant worms. This observation is in line with a previous publication from our lab (Oswald et al., Cell Reports 2018). Furthermore, we observe that the super-resolution maps of wild-type cilia represent a hollow cylindrical distribution, in line with the hollow cylindrical shape of the axoneme. The distribution appears less hollow in mutants lacking kinesin-2, in particular in the proximal segment. Here we speculate that in the cilia of kinesin-II mutant worms the hollow axonemal arrangement is disrupted and the doublet microtubules are more randomly arranged. It is indeed a great idea that EM experts would investigate this further using EM, which we now suggest in our discussion. Overall, we reveal that the axoneme structure in kinesin-II mutant cilia is less organized, even though they have a similar length as wild-type cilia. Rereading our conclusions, we agree that our statements are a bit too strong and could be misinterpreted. We have now toned down the wording, clarified our conclusions and suggested future research directions.

Reviewer #2 (Remarks to the Author): Review comments

In this work, Mitra and colleagues use a single molecule imaging approach to investigate the behaviour of fluorescent-protein tagged IFT motors (kinesin-II, OSM-3, IFT-dynein) and cargo (tubulin) within the proximal-most regions (eg. 1-2 microns encompassing the basal body and transition zone) of sensory cilia in *C. elegans*. Using small-window illumination microscopy (SWIM) in living worms, the authors generated time-lapse recordings of reporter localisations and movements, from which they extracted various types of data (eg, reporter docking sites, pause times, speeds) in WT animals and worms with disrupted IFT motors and transition zones. These data were then used to draw conclusions concerning the spatial and temporal aspects of anterograde IFT train assembly at the ciliary base, and the role of two kinesin 2 motors in this process.

The authors report different ciliary base pause behaviours for the different reporters, with OSM-3 and IFT-dynein pausing for shorter and longer periods, respectively, compared with kinesin-II. Such data is consistent with the motors associating with IFT trains at different stages of their assembly (ie. IFT dynein associates earlier than kinesin-II; OSM-3 associates with trains that are almost fully assembled or fully assembled and starting to move through the TZ). Tubulin hardly pauses at the TZ indicating that it is picked up by fully assembled and moving trains. In terms of docking site, the SWIM imaging shows that whilst kinesin-II and IFT-dynein dock at the proximal part of the TZ, OSM-3 docks over a wider area (and tubulin wider still), consistent with the model of kinesin-II being the initial primary driver of anterograde IFT, after which OSM-3 takes over (as shown previously by this group). For the analyses in IFT motor mutants, the paper shows that kinesin-II docking location, pause time and velocity at the ciliary base are not affected by OSM-3 loss, whereas the reverse is not the case. Specifically, in worms lacking kinesin-II function, OSM-3's pause times increase (at least 1.8 times), its docking location is broader (includes the basal body region), and its initial velocity reduces. IFT-dynein also docks to a wider region of the ciliary base, and shows reduced initial velocities, in kinesin-II mutants. From these data, authors conclude that in the absence of kinesin-II, OSM-3 can take over anterograde IFT transport at the ciliary base, albeit in a less efficient manner. The paper finishes by showing that in worms lacking the TZ regulator *mskr-1*, anterograde train initial velocities are increased and that OSM-3 can dock over a wide region of the TZ; authors conclude that the TZ acts to impede kinesin-II entry into cilia and the initial velocity of anterograde IFT trains.

This paper provides some important insight into the spatial and temporal association of IFT motors and cargo (tubulin) with anterograde IFT trains as they assemble at the ciliary base, and subsequently move through the TZ into the axoneme proper. The single molecule imaging data is of high quality and generally supports the conclusions being made, though not always in all cases (see below)

Major comments

1. Although the paper is well written, the figure layout of the data is not always optimal for following the results as they are presented in the main text. Mostly, the issue was that some result statements required the reader to consider data across multiple figures. For example, in Figure 4 (IFT motor mutants), you have to refer to Figs 1 and 2 to see most of the control (WT) data. Another example is for the results statement: " the super-resolution maps of static OSM-3 and IFT dynein show that the organization at the base is less well defined (Fig. 4G and Extended Data Fig. 6I) compared to wild type (Fig. 3A)...". In this statement, you have to look at the same type of data in 3 figures! There are other examples like this in the paper. A figure should always include the relevant controls for ease of comparison. I can see how this is not so easy to achieve when multiple readouts are being presented for each sample; nonetheless, the authors should make an effort to better include control data in the relevant figures

We agree with the reviewer that it is not ideal to compare data from several different figures for a given finding. We struggled with this and we thank the reviewer for encouraging us to further improve here. We have now substantially rearranged the figures and provided the wild type

conditions (as much as possible) in the figures where we discuss the mutant data. We believe that we have improved the accessibility of the manuscript in this way.

2. Pg 4: “Docking locations show a wider distribution than in wild type, with docking occurring in the TZ as well as at the ciliary base, where OSM-3 takes over the role of absent kinesin-II (Fig. 4B).” To me, the difference between WT and mutant in 2B vs 4B is very slight, if anything at all. This is important to clarify for the conclusions

We did interpret that the docking locations of OSM-3 for the kinesin-II mutant worms are more broadly distributed than for wild-type, with a much higher number of locations in the bin between 0-100 nm. However, we do agree with the reviewer that overall, the difference is subtle. We have toned down our wording, only pointing out that there may be an increase in the number of OSM-3 motors docking at the ciliary base, as indicated by the higher number of events in the first couple of bins.

3. Pg 5: “Second, the super-resolution maps of static OSM-3 and IFT dynein show that the organization at the base is less well defined (Fig. 4G and Extended Data Fig. 6I) compared to wild type (Fig. 3A), with static localizations distributed in a single transversal distribution (instead of two distributions symmetrical around the centre).” In Fig. 4G (left panel), you can still make out 2 distributions although they seem closer together compared to 3A. Whether this can be described as ‘less well defined’ is debatable

The reviewer makes a valid point here and our interpretations were not clearly described. We have now revised this point indicating that in the *kap-1* mutant we also observe two distributions symmetric around the center, although the distributions are closer, additionally providing a **new Figure 4H** showing the histograms of the static localizations corresponding to OSM-3 in wild-type and *kap-1* mutants. We no longer suggest that the organization at the base is less well-defined, instead we suggest that it is altered.

4. Pg 5: “The distribution of the docking locations of OSM-3 in *mksr-1* worms extends much deeper into the TZ and the PS (Fig. 5A,5B), suggesting that OSM-3 can more readily enter the cilium when the TZ is compromised. OSM-3 (Fig. 5F)”. The difference in OSM-3 docking location between WT (334 nm) and *mksr-1* mutant (428 nm) is less than an average of 100 nm; also, the histograms in 3A and 5B don’t show much difference; thus, the notion that docking extends “much deeper into the TZ” is clearly overstated in my opinion.

We agree with the reviewer that we have slightly overstated that OSM-3 docks ‘much’ deeper in the *mksr-1* mutant in comparison to wild-type. We want to point out that the fraction of OSM-3 motors docking deeper into the TZ is higher for the *mksr-1* mutant, which results in the difference in the average docking location. This might indicate that OSM-3 motors find it easier to diffuse further into the cilium. We have now clarified and toned down the way we state this observation and our interpretation.

I would have liked to have seen data for another TZ mutant; for example, the *cep-290* or *mks-5* mutant where TZ structure is more severely affected than the *mksr-1* mutant.

The reviewer makes an excellent suggestion. The question regarding how disruption of the TZ impacts anterograde IFT and entry of ciliary components is an interesting one and our experimental approach would be ideally suited for addressing it. While we do make some striking observations regarding entry of IFT motors in *mksr-1* mutants it would make sense to study the same in mutants where the TZ structure is more severely disrupted, like the *cep-290* or *mks-5* mutants. This would require systematically observing the relevant IFT markers in different TZ mutants. Unfortunately, we do not have the right worm strains readily available to provide some preliminary insights and it would require a much longer time frame to provide the relevant data. Given that the main focus of the study is in describing how IFT components associate with IFT trains to enter the cilia in wild-type cilia, we believe that a detailed investigation in TZ mutants to obtain more nuanced insight is beyond the scope of this manuscript. Having said that, this is definitely a future direction we would like to approach and have discussed this point in more detail in the discussion.

5. Why did the authors not measure the flux rate of IFT particles moving into the axoneme from the TZ, given the observation of increased OSM-3 and kinesin-II initial velocities in the TZ of *mksr-1* mutant? Higher TZ train speeds could lead to more IFT trains entering the cilium per second, or not (either way, this would be very interesting to know).

This is an intriguing point made by the reviewer. However, measuring flux-rate of IFT particles using our imaging approach is not feasible. In our study, the single-molecule imaging conditions are adjusted differently for each IFT component and most of the labelled proteins are photobleached before ciliary entry. Thus, using our approach, it would be challenging to acquire numbers for entry rate accurately, let alone compare between different conditions.

Alternatively, one could get some indications on entry rate by performing FRAP (fluorescent recovery after photobleaching) in cilia. We have now we have performed FRAP experiments to obtain the rate at which OSM-3 enters the cilia of wild-type and *mksr-1* mutant worms (**new Extended Data Fig. 7A-C**). We observed that the initial rate of fluorescence recovery is significantly higher for *mksr-1* mutants. Furthermore, the recovery fraction of *mksr-1* worms is more variable than of wild type. While our results hint towards a higher rate of entry in the mutants, we do not know if this can be attributed to the increased IFT velocity or to the cilia being a more open system, allowing material to enter (and exit) more freely. In the revised manuscript we have discussed this point briefly.

6. Figure 3 claims to “reveal the structure at the proximal part of cilia” and “mapping of ciliary ultrastructure”, concluding “In summary, super-resolution maps provided by single-molecule localizations allow accurate determination of the cylindrical shape of the ciliary proximal part, governed by the axonemal MT doublets.” Whilst this is true, the conclusion does not really yield a lot of new insight, given the many studies in different systems showing IFT trains moving along the periphery of the hollow ciliary axoneme, sandwiched between the ciliary MTs and membrane. Maybe I am missing the point but what is the purpose of providing an “accurate determination of the cylindrical shape of the ciliary proximal part” when this is all known already?

The super-resolution maps for the proximal part of the cilium were a by-product of our single-molecule analysis. Initially we were very excited by the quality of the super-resolution maps we obtained since we thought we could extract the underlying 3D distribution of different IFT-components from their 2D projection maps. In an ideal scenario, this information would not just tell us about the shape of the distribution along the ciliary length but also indicate how far from the axoneme doublets different IFT-components are held. This motivated us to take a deeper dive (see Extended Data Fig. 5 and Extended Data Text associated with Extended Data Fig. 5) and unfortunately, we had to conclude that due to experimental variables, such as localization error and under-sampling of localizations away from focus, we cannot accurately estimate the underlying 3D distributions from the 2D single-molecule localizations. Having said that we still believe that our approach to extract structural information from single-molecule analysis in a living multicellular organism is novel and merits being reported carefully as it potentially will be applicable to other less well characterized systems. Further, we use shape as a parameter to compare structures of wild-type and mutant cilia which becomes relevant when studying the kinesin-II mutant.

Minor comments

1. Introduction: “most ciliary proteins require hitching a ride on an anterograde trains to cross the TZ...” We don’t know how true this is. Certainly some proteins do, but how global and widespread remains to be shown.

We have changed the sentence to indicate that several (and not most) proteins have been shown to require IFT to enter the cilium.

2. Pg 6: “First, we found that the velocities of all IFT components studied are significantly lower in the densely crowded TZ than further in the cilium, as has been observed before^{8,27}. Furthermore, in *mkrs-1* worms with disrupted TZ, we observed that kinesin-II and OSM-3 move faster in the TZ and initial part of the PS than in wildtype.” That the TZ supports slower IFT speeds, regulated by TZ complexes, has been shown before in *C. elegans* in <https://pubmed.ncbi.nlm.nih.gov/26392567/>. This paper must be cited.

This is indeed a highly relevant paper and we missed it while writing the original manuscript. We now refer to in the discussion.

Reviewer #3 (Remarks to the Author):

Summary

The intraflagellar transport (IFT) system to ferry cargoes into the cilia which is carried out by multiple proteins including kinesin, OSM, dynein and tubulin, working in tandem across the packed transition zones. In nematodes, this involves slower navigation by kinesin-II in the anterograde direction across the transition zone and faster transport by OSM-3 to the apex of

the cilia. Mitra et al in this study make use of a novel imaging tool – small-window illumination microscopy (SWIM) to increase the signal-to-noise ratio and to decrease the out of focus autofluorescence so as to carry out single molecule tracking of the IFT motors in live worms. They carried out their experiments in the PHA and PHB phasmid neurons of the worms. Their tracking experiments were combined with simulations and nanoscale localization experiments. In addition, the authors provided data on wild-type and kinesin-deficient worms, detailing their importance in the organisation, and directed transport of cargoes within the cilia.

Major Comments

- The authors apply the novel imaging tool – SWIM to obtain the movement of IFT associated proteins in vivo. The authors should confirm the validity of this narrow-window approach in at least a different neuron within the worms. Is its use limited to only the thin and transparent tip, or could it be expanded to image other cilia for example in the chemosensory neurons in the head of *C. elegans*?

We refer to an earlier publication from our group (Mitra et al., *Optics Communications* 2023) for all details, physical explanation, and validation of our SWIM illumination scheme.

The general idea of SWIM is also applicable for imaging amphid sensilla in the head of worms. However, there are several technical challenges that make quantitative single-molecule analysis, as performed in phasmid sensilla, very difficult: (1) Amphid sensilla consist of a bundle of 12 cilia (variable in structure and length) and it is difficult to distinguish between them consistently such that we observe the same cilium in different worms. In contrast phasmid sensilla consist of only one cilium pair and the structure of these cilia as well as IFT appears to be identical. (2) SWIM requires continuous imaging of IFT in the same focal plane. The tail of the worm lies flat on the surface and it is often possible to have the entire ciliary pair in a single focal plane of imaging. In the case of amphids, it is impossible to obtain the ciliary base of all cilia in the same focal plane. A confocal imaging approach scanning swiftly in the z-direction would be more suitable for imaging the amphids. (3) Even though the worms are sedated using levamisole during imaging, the head twitches much more frequently during imaging making it impossible to perform quantitative single-molecule analysis on amphid cilia.

Having said that, we have now imaged IFT-dynein in amphid cilia and now provide the imaged single-molecule data as a **new Supplementary Movie 6**, also referring it in the discussion. We indeed observe that the worm head twitches regularly and only a few of the cilia in the amphid bundle have their ciliary base in focus, which highlights why we have performed the study focusing only on the PHA/PHB cilia pair at the tail. Qualitatively, however, we do observe that IFT dynein is diffusive in the dendrites with individual motors occasionally docking at the ciliary base, pausing for a while, before moving into the cilia. This highlights that the mechanism of ciliary entry of IFT motors we describe in this study, based on our investigation on PHA/PHB cilia pair, might be more generally applicable to other cilia.

- The tagging of the ciliary proteins with eGFP suggests that the SWIM method might be imaging like FRAP (that is, bulk protein movement imaging rather than individual protein tracking). With the use of eGFP, even in the narrow window imaging approach, the authors cannot rule out that bright spots are representative of individual proteins separated by the distance of the diffraction limit – about 250nm.

We note that we have published many papers on single-eGFP imaging in live *C. elegans* before, providing extensive quantification and confirmation that we indeed observe single fluorophores (Prevo et al., Nature Cell Biology 2015, Mijalkovic et al., Nature Communications 2017, Mijalkovic et al., Cell Reports 2018, van Krugten et al., Communications Biology 2022, etc.). In SWIM we use a slightly altered excitation scheme, but many of the other microscope components (objective, filters, detector) are identical, which convinces us that we are imaging single molecules using SWIM (see detailed reply to the first point from Reviewer 1). To further convince the reviewer, in the revised manuscript, we have now imaged IFT-dynein, kinesin-II and OSM-3 at a ~5-10x faster rate. In these faster acquisitions we clearly see individual molecules diffusing in the PCMC (whenever they diffuse primarily in the plane of focus) before docking onto a paused or moving train. We now provide new **Extended Data Fig. 4 and Supplementary Movie 3** in the manuscript making the point that we observe individual molecules in the acquired movies.

Further, in our analysis, we are careful to select only individual molecules and ignore events when the ciliary base is busy, i.e. when puncta become too close to resolve. This selection is manual and it is plausible that occasionally we are selecting more than one fluorescing protein binding to the same train within a short time interval (difficult to discern at our imaging speed). This however would only be an issue for a small percentage of tracks and our overall observations and analysis would still hold true. We have now addressed this caveat in the methods section where we discuss how we select single-molecule tracks.

Therefore, the authors should verify their data with either photoactivatable-GFP, photoconvertible mEOS3 or Halo tag to verify single protein data acquisition.

Using photoactivatable or photoconvertible probes will indiscriminately activate proteins throughout the cilium and we have used this approach to observe single-molecule IFT dynamics in previous studies (Prevo et al., Nature Cell Biology 2015 and Zhang et al., PNAS 2020). Using that approach, we have shown that we can observe the dynamics of individual molecules, but we do not specifically observe the molecules that freshly enter the cilium. Our current approach, SWIM, allows us to continuously image single-molecules entering the small field of view that is illuminated by the excitation laser and, in our opinion, is the ideal way to study single-molecule dynamics of IFT proteins entering the cilium.

- The authors should confirm that the differences seen in docking locations, and pause times between kinesin-II, dynein, and OSM-3 is not due to differences in the levels of expressions of the proteins. If double copy numbers of kinesin-II were present, does this in anyway reduce its mobility?

In our study, all strains were made using the MosSCI method (Frøkjær-Jensen et al., Nature Genetics 2008), that inserts a single copy of a transgene into a defined site. We inserted single-copy transgenes encoding for fluorescently labelled IFT components in the corresponding null mutant worm. Thus, we have single copy number for each IFT-component we visualize. Further, all the worm strains used in this study have been used in previous publications from our laboratory and in all cases, we find that the fusion protein rescues normal ciliary function in the respective null mutants.

Perhaps, the reviewer is suggesting that inherently different IFT components have different endogenous levels of expression with respect to each other, for instance, the overall

concentration of OSM-3 is much higher than kinesin-II, which results in the differences seen in docking locations and pause times. This is indeed something worth thinking about and it might be interesting to explore how binding dynamics during ciliary entry is altered upon overexpression. However, given that different IFT components have different train binding affinities and binding sites, we believe the effect of overexpression would be different for different IFT component. Thus, we believe showing docking location distributions and pause times for kinesin-II upon overexpression (as suggested by the reviewer) does not really reveal much about the differences observed for ciliary entry of different components.

- The authors should plot the area under the curve for the velocity plotted for the proteins (kinesin-II, dynein, tubulin, and OSM-3) and determine whether they are statistically different or not. This should be applied to other figures as well.

In the figures we plot the velocity along the cilia length. For the plots we indicate the average binned velocity and the error (obtained from bootstrapping). Wherever the errors for compared datasets do not overlap, they are statistically significant. We think using the area under the curve for such plots to perform a significance test is not meaningful. To determine whether two velocity datasets are statistically different along the length of the cilia, one would have to compare the velocity data for every bin (100 nm wide) along the cilia length. This, we think is unnecessary since we directly see how different or alike the velocity distributions are.

- How are the 3D-distributions of tubulin on the axoneme, altered in kinesin-deficient and SOM-3 worms?

We have addressed this for kinesin-II mutants in a previous study from our laboratory (Oswald et al., Cell Reports 2018). In this study it was also observed that the bulge in the axonemal structure at the proximal segment of the cilia is missing in kinesin-II mutant cilia. We have mentioned the observations made in this study in our manuscript. For OSM-3 mutants, the structure at the proximal part of the cilia is similar to wild-type cilia. Hence, we do not find it necessary to visualize tubulin in cilia of these mutants.

- Do drug inhibitors like monastrol of the IFT proteins like kinesin, similarly alter the super-resolution map?

Inhibiting motor action of a specific motor by chemical, optogenetics or other means in wild-type cilia and then observing how IFT and cilia structure is altered in response would be an exciting future direction. Drugs, specifically targeting a single protein would be an approach worth exploring. However, we are not aware of any drug that specifically inhibits kinesin-II or OSM-3. Ciliobrevin is a known inhibitor of dynein but it would also inhibit cytoplasmic dynein along with IFT-dynein which may have other complications elsewhere within the neuron. Hence, we do not want to pursue this direction for now.

Minor Comments

- The authors should correct for the repetition of 'that' in the sentence: 'From the single-molecule tracks, we calculated that the average velocity and observed that it increases...'

We have changed the sentence as suggested.

- How did the authors account for micro- or nano-movement of the worm tail during data acquisition?

If the worm tail makes micro-movements during our imaging, it is clearly visible and we have ignored such worms from our analysis. We now state this in the methods section. Nano-movements would be trickier to account for and could have a minor impact on the numbers we acquire from our analysis. To account for this one would require having fiducial markers, that do not bleach, within the body of the worm being imaged. We have not figured out a way to do that robustly. We now mention this caveat in the revised manuscript.

REVIEWERS' COMMENTS

Reviewer #1 (Remarks to the Author):

The authors have addressed most of my concerns with new experimental evidence, additional data analysis, and in-text revisions. I have only one minor concern before I can recommend publication.

The authors claim that OSM3 and tubulin instantaneously enter the cilium and hop on IFT trains already moving in the cilium. While this is their interpretation, I recommend the authors stick more closely to the experimental observations in the Results section. I think their results show that dynein-2 shows long pauses, and kinesin-II shows brief pauses. However, they could not detect pausing events for OSM3 and tubulin. These observations suggest that OSM3 and tubulin bind to IFT after dynein and kinesin-II, but these results do not clearly show that they only bind to moving IFT trains. I recommend replacing “instantaneously” with a more suitable word in the title of that section, and reporting what the authors were and were not able to observe (i.e. no clear pausing for OSM3 and tubulin) before they interpret their observation.

Reviewer #2 (Remarks to the Author):

The authors have performed a good job in addressing my comments.

Reviewer #3 (Remarks to the Author):

The authors have sufficiently addressed all my previous questions and comments.

Response to Reviewer's comments

We would like to thank the reviewers for their approval of our revised work.

Reviewer #1 (Remarks to the Author):

The authors have addressed most of my concerns with new experimental evidence, additional data analysis, and in-text revisions. I have only one minor concern before I can recommend publication.

The authors claim that OSM3 and tubulin instantaneously enter the cilium and hop on IFT trains already moving in the cilium. While this is their interpretation, I recommend the authors stick more closely to the experimental observations in the Results section. I think their results show that dynein-2 shows long pauses, and kinesin-II shows brief pauses. However, they could not detect pausing events for OSM3 and tubulin. These observations suggest that OSM3 and tubulin bind to IFT after dynein and kinesin-II, but these results do not clearly show that they only bind to moving IFT trains. I recommend replacing "instantaneously" with a more suitable word in the title of that section, and reporting what the authors were and were not able to observe (i.e. no clear pausing for OSM3 and tubulin) before they interpret their observation.

We have now replaced the word 'instantaneously' in our manuscript, which we agree was our interpretation and not the observation itself.

Reviewer #2 (Remarks to the Author):

The authors have performed a good job in addressing my comments.

Reviewer #3 (Remarks to the Author):

The authors have sufficiently addressed all my previous questions and comments.